# Decoding technical multi-promoted ammonia synthesis catalysts

Luis Sandoval-Díaz [1] ✉, Raoul Blume[2], Kassiogé Dembélé [1], Jan Folke[2], Maxime Boniface[1], Frank Girgsdies [1], Adnan Hammud [1], Zahra Gheisari[1], Danail Ivanov[1], René Eckert[3], Stephan Reitmeier[3], Andreas Reitzmann[3], Robert Schlögl[4], Beatriz Roldan Cuenya [4], Holger Ruland [2], Axel Knop-Gericke[2] & Thomas Lunkenbein [1] ✉

Ammonia is industrially produced by the Haber-Bosch process over a fused, multi-promoted iron-based catalyst. Current knowledge about the reaction has been derived from model systems of reduced structural complexity, impeding any clear-cut structure-activity correlation relevant for the industrial counterpart. Here, we unveil the structural evolution of complex, technical, multi-promoted ammonia synthesis catalysts by operando scanning electron microscopy and near-ambient pressure X-ray photoelectron spectroscopy. We highlight that the activation is the critical step in which the catalyst is formed and decode the pivotal role of the promoters. We discover that the active structure consists of a nanodispersion of Fe covered by mobile K-containing adsorbates, so called "ammonia K". The porous catalyst is stabilized by mineral cementitious phases containing oxides of Al, Si, Ca, and Fe. The synergism between the different promoters contributes simultaneously to the structural stability, hierarchical architecture, catalytic activity, and poisoning resistance. The confluence of these aspects is the key for the superior performance of technical catalyst formulations.

The Haber-Bosch process for producing ammonia from the elements at industrial scale was introduced over 110 years ago. Since then, innovations in this field have been minimal[1–6]. Although contemporary practices employ wüstite-based precursors, which are more effective than the earlier magnetite-based materials[4], and a broader array of promoters has been implemented[7–9], the core principles of industrial ammonia synthesis have remained as they were originally conceived. These principles involve a recirculating reaction stream that is transformed over an iron-based catalyst produced through melt synthesis of oxidic precursors. This enduring technological consistency contrasts with the deepening scientific understanding of the reaction, making ammonia synthesis a unique example in heterogeneous catalysis[10].

Through experimentation on single crystal model catalysts, a detailed understanding of the involved molecular mechanisms has been achieved, revealing insights into the rate-limiting step[11], the structural dynamics[12], and the sensitivity of the reaction to the iron surface structure[11,13,14]. However, while these models have been successful in elucidating molecular details, they fall short in capturing the complex interactions existing in the actual technical catalyst[15]. These interactions can lead to additional dynamics, including phase transformations[16] and structural fluctuations[14] that are distinctive of the technical formulation exposed to the reaction environment. Furthermore, model system studies often focus on idealized processes with homogeneous active sites that lack mass-energy transport limitations and deactivation. This view cannot be harmonized with the

[1]Department of Inorganic Chemistry, Fritz-Haber-Institute of the Max-Planck-Society, Berlin, Germany. [2]Department of Heterogeneous Reactions, Max Planck Institute for Chemical Energy Conversion, Mülheim an der Ruhr, Germany. [3]Clariant Produkte (Deutschland) GmbH, Heufeld, Germany. [4]Department of Interface Science, Fritz-Haber-Institute of the Max-Planck-Society, Berlin, Germany. ✉e-mail: lesandovaldi@fhi-berlin.mpg.de; lunkenbein@fhi-berlin.mpg.de

realities of heterogeneous catalysis, leaving gaps in our understanding of the actual process.

One significant gap pertains to the active iron phase known as "ammonia iron". In-depth characterization has shown that ammonia-iron possesses a nanometric layered structure and hierarchical porosity[10,15,17]. While its principal chemical properties, such as electronic state, average crystal structure, and coordination, only slightly differ from α-Fe, ammonia iron sustains ammonia synthesis for 7–10 years under industrial conditions[17]. Hence, its performance cannot be attributed merely to a specific orientation of the α-Fe crystal[10,15,18]. In addition, the presence of promoters, such as $K_2O$, has profound impact on the activity and structural sensitivity, whereas other additives, such as CaO and $Al_2O_3$ are crucial for the structural integrity[16,18,19]. It is evident that the interplay of all the components of the multi-promoted catalyst is essential for the active catalyst configuration, highlighting the necessity for comprehensive approaches to understand the complexities inherent to the reaction[20]. This complexity was acknowledged early on by Fritz Haber, who wrote to Alwin Mittasch that the function of the promoters was "enigmatic"[4] in light of the poor performance of unpromoted iron.

In this study, we examine the intricate interactions between the components of a technical multi-promoted ammonia synthesis catalyst using operando scanning electron microscopy (OSEM)[21] and near-ambient pressure X-ray photoelectron spectroscopy (NAP-XPS)[22,23]. Our goal is to elucidate the structural and compositional changes of the catalyst during activation and during ammonia synthesis. Our findings reveal that the active configuration consists of a hierarchical porous nanodispersion of metallic iron covered by mobile potassium-containing entities that we term "ammonia K". The research demonstrates that the activation process is the truly critical step where the active catalyst configuration is formed through an interplay of influences induced by the promoter phases. The orchestrated action of the promoters includes a controlled reduction kinetics of the wüstite precursor, the formation of cementitious materials imparting structural stability, and the increase of the local alkalinity. These effects account for the differences between ammonia iron and conventional α-Fe, giving rise to the technical catalyst's superior durability and activity.

## Results
### Catalyst overview
A technical, multi-promoted, wüstite-based ammonia synthesis catalyst was used in this study. The most relevant promoters of this formulation are $Al_2O_3$, CaO, $SiO_2$, and $K_2O$[16]. Alumina is considered as a structural promoter that increases the catalyst surface area. Its presence has also been related to a solid-state reaction with $FeO_x$, leading to $FeAl_2O_4$[18] that modulates the rate of reduction during activation[24]. CaO also increases the surface area as well as the resistance against gas impurities[19], and $K_2O$ is considered as an electronic promoter that enhances the rate of ammonia production and reduces the propensity to self-poisoning[19,25,26]. The synthesis and structural characteristics of the technical, multi-promoted ammonia synthesis catalyst are discussed in detail elsewhere[27].

An SEM survey of the initial catalyst (Fig. 1a–c) shows a complex morphology at different length scales. At the micrometer scale (Fig. 1a), the catalyst exhibits discrete and fused granules. These granules are seemingly composed of iron-rich oxides combined with oxidic promoters (Si, Al, Ca, K, Supplementary Fig. 1). In contrast, the interstitial material holding the granules together is depleted in iron and enriched in the promoter elements (Si, Al, Ca, Supplementary Fig. 1). At higher magnification (Fig. 1b, c), the Fe-rich granule appears highly heterogeneous, exhibiting a rough surface decorated by arbitrarily distributed particles extending from the nanoscale to several microns. The substrate material of the granule has a porous structure (Fig. 1c). Hence, the technical catalyst is a multi-scale heterogeneous and structurally complex entity.

The morphological characteristics of the initial catalyst contrast with the post-reaction material (Fig. 1d). After 96 h of ammonia synthesis at 90 bar, the previously described surface features are unrecognizable and new structures have been formed. We identify crust-like, needle-like, and platelet-like materials that coexist on the catalyst surface. An X-ray diffraction characterization (Supplementary Fig. 2) shows that the wüstite material of the initial catalyst was transformed into metallic iron after the reaction. In addition, the structural types of gibbsite, melilite, and tricalcium aluminate ($C_3A$) were detected as crystalline promoter phases at the initial catalyst. After the reaction, the gibbsite phase seemingly disappeared, leaving only the melilite and the tricalcium aluminate in the spent catalyst.

### Catalytic activation by operando SEM
Thus, we conducted OSEM experiments to explore in detail the structural transformations leading to the final catalyst. Figure 2 shows the heating protocol of the experiment and the evolution of the products at m/z = 18 ($H_2O$), 17 (OH, $NH_3$), 16 (O, $NH_2$), and 15 (NH), respectively. The water signal decreased during the first 120 h of the experiment, except for three peaks at time on stream (TOS) = -5, -40, and -80 h. We attribute the first peak to the desorption of moisture from the reactor. During this time, the signals at m/z = 15 and 16 were steady and very low. The second water peak occurred in the interval between 120 and 295 °C. This peak can be related to the reduction of the catalyst, accompanied by the activation of ammonia synthesis (pink area). Subsequently, the production of ammonia increased, while the evolution of water decreased until the end of the temperature ramp, except for a third water peak that we attribute to the completion of the catalyst reduction. Afterwards, the system stabilized during the subsequent isothermal regime at 500 °C, where negligible water formation and a very slight activation of ammonia production (TOS = 74.5–300 h) were detected. For comparison, we conducted the activation procedure ex situ at varying pressures (1–30 bar) and found a similar behavior of the reaction traces (Supplementary Figs. 3–5 and Supplementary Note 1). We also calibrated the QMS signal at m/z = 15 for semiquantitative determination of the ammonia conversion from the OSEM data. As shown in Supplementary Table 1, the estimated conversion rate was below the values reported for iron single crystals under high-pressure conditions[28].

Figure 3 illustrates the evolution of the catalyst surface during the operando experiment. The SEM images in Fig. 3a–c show the exsolution of nanometric material from the porous surface of the Fe-rich granule during the temperature ramp. At a temperature of -377 °C, roundish nanometric particles are already observed at the granule surface (Fig. 3a). Upon heating, the size and abundance of these particles increased (Fig. 3b, c). A visualization of this process is presented in Supplementary Movies 1 and 2. Supplementary Movie 1 shows the catalyst surface during the temperature ramp, resulting in a drifting field of view due to the thermal expansion of the solid and the occurrence of chemical transformations. Supplementary Movie 2 shows an aligned region of interest (ROI) extracted from Supplementary Movie 1 in the interval between 340 and 450 °C, demonstrating the formation and growth of the nanoparticles at the catalyst surface. The onset of this exsolution seemingly correlated with the catalyst reduction and activation of ammonia production (pink area of Fig. 2).

The morphology of the catalyst surface changed further during the subsequent isothermal treatment at 500 °C (Fig. 3d–i and Supplementary Movie 3). At TOS = 76.5 h, roundish nanoparticles of the exsolving material were still present. Their abundance, surface coverage, and sizes continually increased during the first 24 h of the isothermal treatment. At this stage, the changes can be best described as the formation of additional nanomaterial and the texturization of the surface, i.e., changes in the porosity of the catalytic substrate at the Fe-rich granule. The footage in this regime suggests that the roundish nanoparticles that were formed at the early stages served as nuclei for

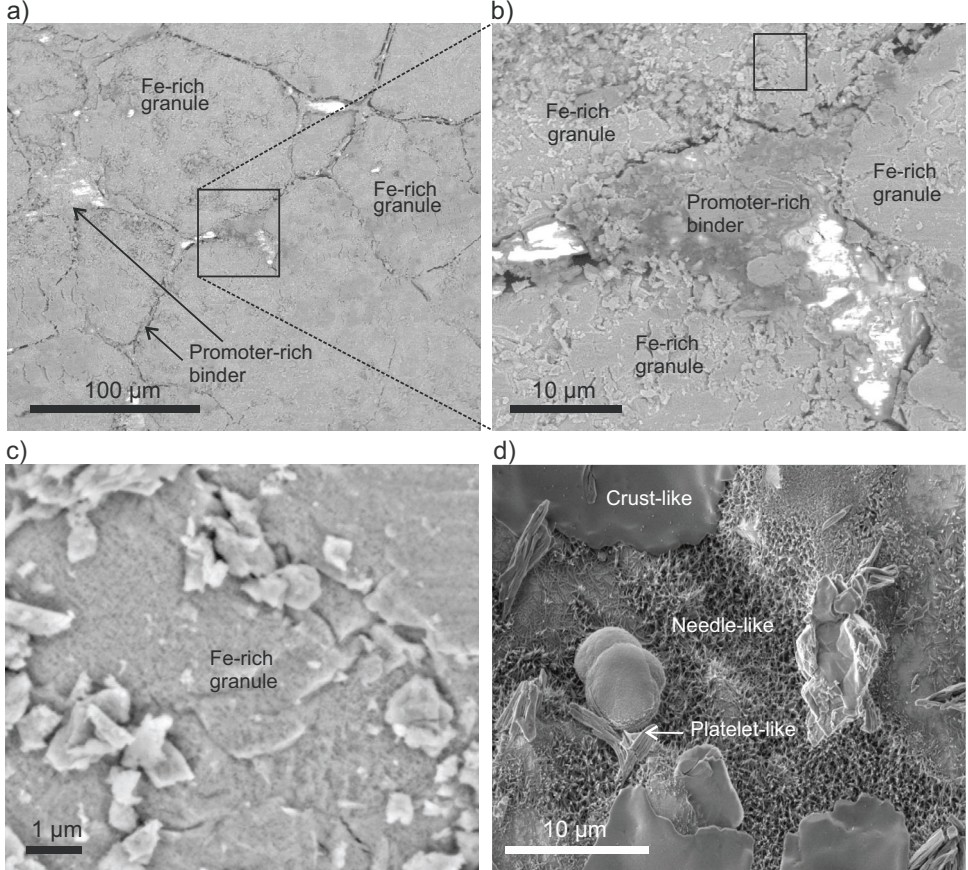

**Fig. 1 | SEM survey of the surface morphology of a technical multi-promoted ammonia synthesis catalyst. a** The overview image of the surface structure of the initial catalyst shows a brick wall architecture with iron-rich granules held together by a continuum of promoter material prior to catalytic reaction. **b** SEM image of the region of interest highlighted in (**a**) displaying the complex structure in a confluence point of several iron-rich granules. **c** The close-up look on an iron-rich granule from the region of interest highlighted in (**b**) denotes the surface morphology covered by a heterogeneously distributed material and a maze-like porous structure of the substrate. **d** SEM image of the surface of the spent catalyst after 96 h of ammonia synthesis at 90 bar showing platelet, needle-like, and crust-like materials. Conditions of acquisition of (**a**–**c**): 22 Pa, $H_2:N_2:Ar = 3:1:0.1$.

subsequent material growth. After TOS = 146.5 h, the size and coverage of these particles increased, and a platelet-like morphology was obtained (Fig. 3f).

We imaged different regions of the catalyst at various magnifications to assess how representative these observations are. The overview image in Fig. 3g shows that most of the iron-rich granules detected at the beginning of the experiment were covered by the platelets at TOS = 253 h. A closer inspection (Fig. 3h) shows the complex microstructure of the platelets. Individual platelets had sizes ranging from 50 to 200 nm in length and 20 to 80 nm in thickness. Furthermore, the particles grew preferentially in the direction perpendicular to the external surface (Fig. 3i). The approximate height of these platelets after TOS = 253 h ranged between 200 and 600 nm.

After TOS = 272 h (Fig. 4a), the catalyst underwent an important morphological change (Fig. 4b, c and Supplementary Movie 4). While part of the platelet material was still growing, other areas started to disintegrate, giving rise to disordered and needle-like phases. This transformation propagated from the collapsed material to the vicinal platelets, akin to a domino effect. After a few hours, the platelets had almost completely transformed into this disordered phase, giving rise to patches of a crust-like morphology. Notably, this structural transformation did not bring any appreciable changes in the catalytic conversion to ammonia (blue region of Fig. 2). Hence, collapsed platelets, crust-like material, and areas of intricate acicular morphology remained on the catalyst surface until the end of the run.

We repeated the experiment under the same conditions on a fresh catalyst aliquot to investigate the reproducibility of the observed dynamics and to analyze the structure and composition of the platelet-like phase. The catalyst surface in the second run exhibited a similar behavior, confirming the distinctive morphological evolution during the activation process (Fig. 4d–f and Supplementary Movie 5). We quenched the reaction before achieving the regime of disintegration of the segregated phases by controlled reduction of the reactor temperature (TOS = 235 h). Afterwards, the spent catalyst was transferred in an inert environment for ex situ characterizations of the local structure.

**Post-reaction catalyst characterization**

The images presented in Supplementary Fig. 6a, b were acquired during the preparation and lift-out of a thinned cross-section (lamella) of the spent catalyst material from the second OSEM experiment, using Focused Ion Beam (FIB) milling. The images exemplify the segregation of platelet material into the external surface of the catalyst, as observed during the OSEM experiments. Furthermore, Supplementary Fig. 6b reveals the porous nature of the catalytic substrate. The formation of an open porous system is consistent with previous gas adsorption measurements, showing a surface area increase from approximately $2 \, m^2g^{-1}$ to $15\text{–}19 \, m^2g^{-1}$ following reductive activation in multi-promoted systems[16,29,30].

The HAADF-STEM (Fig. 5a) and the low-resolution BF-TEM (Supplementary Fig. 6c, d) images of the cross-section corroborate the

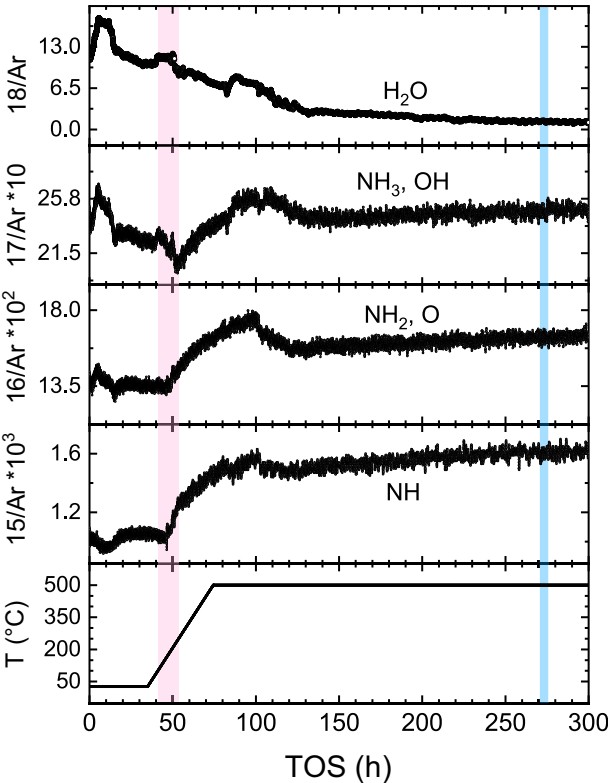

**Fig. 2 | Time series of temperature protocol and products during the OSEM experiment.** The time series shows the heating protocol and the ion currents of NH, NH₂, NH₃, and H₂O (normalized to Ar) with major contributions at m/z = 15, 16, 17, and 18, respectively. The ionic currents at m/z = 15, 16, and 17 were multiplied by 500, 100, and 10, respectively. The pink region marks the onset of the catalytic activation with simultaneous catalyst reduction. The blue region marks the phase transformation of the surface into a disordered material. Conditions of acquisition: 22 Pa, H₂:N₂:Ar = 3:1:0.1. Heating ramp: 12 Kh⁻¹.

porous nature of the catalytic substrate. Furthermore, as indicated in Fig. 5b–g, the iron was homogeneously distributed across the catalyst, while the promoter elements were found to dominate the composition of the platelets at the outermost regions of the structure. The relative variations of elemental compositions (Fig. 5h) along the plotline (arrow region in Fig. 5a) show an increasing O/Fe ratio in the platelets compared to the substrate. The promoters-to-Fe ratios exhibit similar behaviors, increasing once the plotline reaches the position of the platelets and indicating an enrichment of the promoters at the catalyst surface correlated to the position of the platelets. Notably, the Al/Si and Ca/Si ratios remained unaltered along the plotline despite the individual compositional variations with respect to Fe. This observation suggests that Al, Ca, and Si formed a homogeneous mixture or a compound throughout the catalyst[31].

The high-resolution TEM micrograph of the platelets (Fig. 5i) implies a stacked structure that grew orthogonal to the surface. This ordered structure was found to be covered by an amorphous rim. Complementarily, the high-resolution TEM images of the substrate material (Fig. 5j, k) reveal the Fe grains, which are characterized by a heterogeneous size distribution and were found to be separated by an amorphous material, giving rise to the porous arrangement. In addition, surface steps were observed at locations uncovered by the amorphous material (Fig. 5k, black arrows).

### Chemical species by in situ NAP-XPS
Next, we assessed the evolution of the surface composition by NAP-XPS measurements. NAP-XPS data of Fe, N, K, Al, Si, and Ca are

presented in Fig. 6, Supplementary Figs. 7 and 8, and Supplementary Table 2. At 250 °C, the catalyst surface was dominated by FeO$_x$ species (Fig. 6a). At 500 °C, metallic iron was mainly observed, as evinced by the shift of the binding energies (BE) from 708 to 714 eV to BE = 706.9 eV. A small peak at BE = 707.7 eV was detected under these conditions, which can be attributed to iron nitrides[32]. Furthermore, a tiny amount of refractory iron oxides could still be detected. The changes in the oxidation state of iron were partially reversible after cooling the catalyst to room temperature in the reaction mixture, indicating that the reduction of the bulk may have not been completed during the treatment in the NAP-XPS experiment. Nonetheless, some metallic iron remained after the temperature cycle. We attribute this observation to the short duration of the treatment in the NAP-XPS experiment (126 h in total) compared to the OSEM (300 h).

A reversible behavior was also observed in the N1s spectra (Fig. 6b and Supplementary Fig. 7b). Signal fittings indicate the presence of at least three different nitrogen species corresponding to atomic N (BE = 396.9 eV), γ′ (BE = 395.9 eV, Fe₂N), and ε (BE = 397.9 eV, Fe₃N) iron nitrides[33]. Notably, the peak associated to Fe₂N was most prominent at 250 °C, when the sample was just starting to be reduced. The most prominent signal at 500 °C is attributed to atomic N with a smaller participation of the iron nitrides. After cooling, the signal assigned to γ′ nitride became dominant again, the atomic N contribution was reduced, and NH₂ and NH₃ species were detected at the catalyst surface.

For K species (Fig. 6c), the 2p$_{3/2}$ contributions indicate a mixture of mainly K₂O (BE = 293.2 eV) and KOH (BE = 293.8 eV) at 250 °C. At 500 °C, emergence of a peak at BE = 294.75 eV was observed. This signal, which we refer to as " ammonia K", may indicate the formation of metallic K[34] or undercoordinated isolated surface K⁺ species[35]. In addition, a small signal at 296.0 eV also appeared. This last signal with upshifted BE indicates an inefficient dissipation of charging produced by the X-ray excitation. The changes were mostly reversible after cooling the catalyst to room temperature, although part of the in situ-formed "ammonia K" still survived.

In contrast, the thermal behavior of the promoters Al, Si and Ca was found to be mainly irreversible upon the temperature cycle (Fig. 6d–f and Supplementary Fig. 8). At 250 °C, the spectra of Al and Si exhibited signals at low BE that can be attributed to small units of aluminate/aluminum hydroxides and silicates, respectively[36,37]. Under these conditions, the calcium material was composed of a mixture of CaO, Ca(OH)₂, and (silico)aluminates[38]. At 500 °C, the signals corresponding to aluminum hydroxides decreased, while the formation of polymeric silicoaluminates and silica became dominant. In line, the initial CaO/Ca(OH)₂ mixture was transformed into calcium silicates[39] and calcium silicoaluminates[36].

To qualitatively assess the surface composition, we integrated the intensities of the XPS signals of the promoters as a function of temperature (Fig. 7). The compilation of data represents acquisitions at different experiments on different catalyst aliquots. The data show strong variations due to the local structural diversity. However, the trend indicates that there was an enrichment of Al, Si, and Ca at the surface with increasing temperatures, while the opposite behavior was observed for the K promoter.

## Discussion
The catalyst contains two types of promoter materials. One type is mixed with the FeO$_x$ granule, while the other type forms a phase at the interstices of the granules. This phase segregation occurs during the solidification of the molten mixture of the precursor oxides due to differences in solubilities[4,31]. The promoter phases connecting the Fe-rich granules can form compounds that are similar to glasses or minerals[15], including silicoaluminates, silicates, silicoferrites, among others. The role of these phases for preserving the catalyst structural integrity is obvious (Fig. 1a). Complementarily, the promoters found at

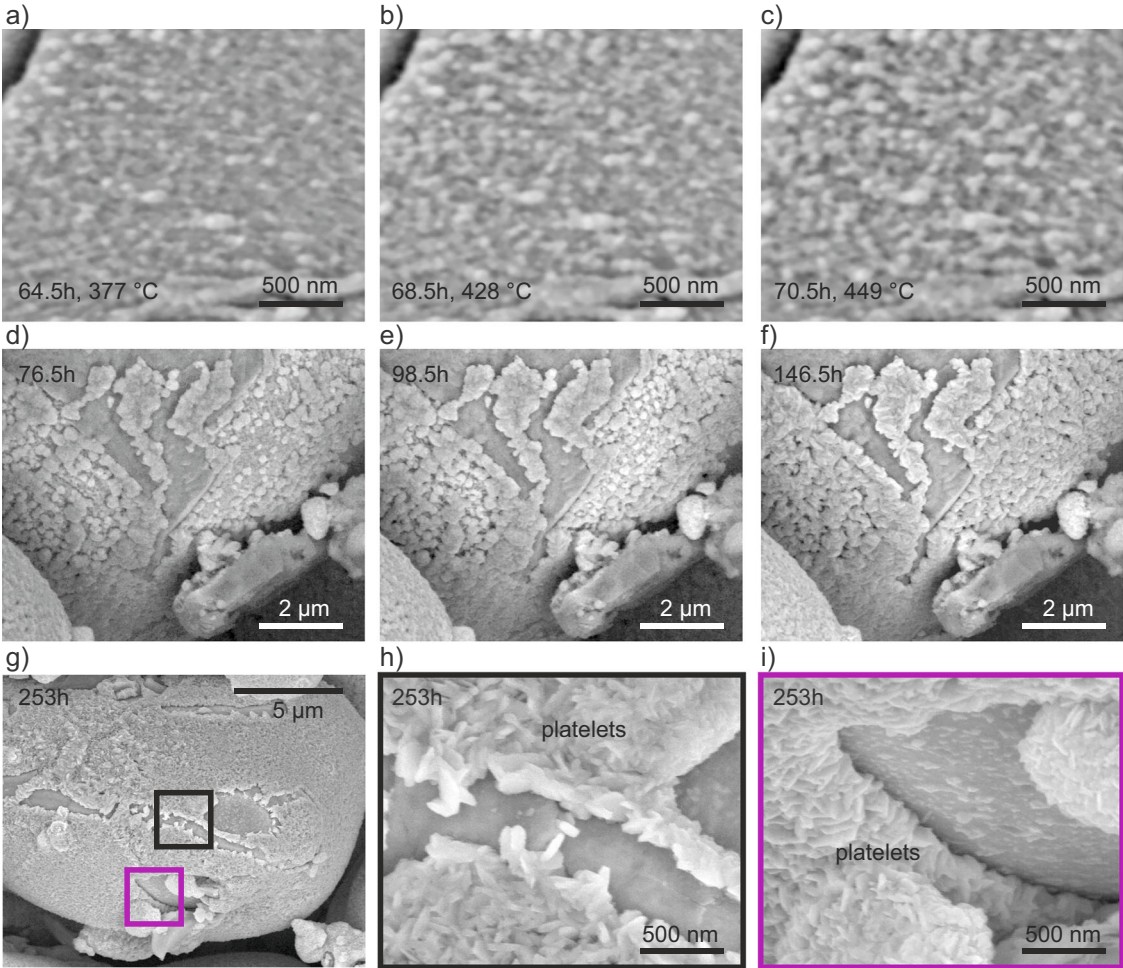

**Fig. 3 | Consecutive in situ SEM images of the catalyst surface at iron-rich granules during heating and under reaction conditions. a–c** The evolution of the catalyst surface during the temperature ramp. **a** The segregation of material is already detectable at 377 °C at TOS = 64.5 h in the form of bright roundish nanoparticles on the catalyst surface. **b** The segregation of material continued at 428 °C and TOS = 68.8 h. **c** The treatment induced growth of the segregated phase at 449 °C and TOS = 70.5 h. Images extracted from the Supplementary Movies 1 and 2. **d–i** Evolution of the catalyst surface during the isothermal treatment at 500 °C showing platelet formation. Images in (**d–f**) were extracted from Supplementary Movie 3. **g** Overview image showing the external surface coverage by the segregated phase after TOS = 253 h. **h, i** Close-up views of the regions of interest marked in (**g**) showing the platelet-like morphology of the segregated material. Conditions of acquisition: 22 Pa, $H_2:N_2:Ar = 3:1:0.1$.

the Fe-rich granules make ammonia iron distinct from pure α-Fe. Starting with the $FeO_x$ precursor, the structure of the resulting metallic phase suggests that the reduction must have been different to a direct reduction of bulk wüstite by either disproportionation or by core-shell mechanisms. These mechanisms would produce dense metallic overlayers and disordered material over partially reduced $FeO_x$[16,17]. We did not observe this kind of structures. Instead, the ammonia iron exhibited a well-crystallized structure (Fig. 5j, k and Supplementary Fig. 2) that must have been achieved by a homogeneous propagation of the reduction front.

Consequently, the reduction must have been kinetically controlled by the promoter phases, for instance, by mediating the mass transport of $Fe^{2+}$. The observed structure suggests that it is necessary to give sufficient time for the migration of $Fe^{2+}$ cations to relocate and order in the metallic phase. It is known that $Fe^{2+}$ can form substitutional compounds with the promoter oxides, such as $FeAl_2O_4$, iron melilite, calcium and aluminium silicoferrites[27,40,41], and solid solutions with other phases such as tricalcium silicate, tricalcium aluminate, and calcium silicate. $Fe^{2+}$ in these phases can migrate more effectively than in wüstite due to higher ion mobilities and preferential gliding planes[17]. For instance, Mössbauer spectroscopy experiments have confirmed that $Fe^{2+}$ can substitute several cations in defective oxide

structures[42,43]. These effects enhance the homogeneous reduction of the $FeO_x$ since the local concentration of reactive $Fe^{2+}$ can be continually resupplied from the phases formed with the promoters. In addition, the relocation of matter supports the formation of porous architectures while preserving the crystalline phase.

With the ongoing reduction, the promoter and the metallic phases become immiscible. Hence, the promoters segregate at the interstices of the metallic crystallites and into the external surface[44] (Figs. 3a–i, 5j, k, and 6 and Supplementary Movies 1–3). Compositional changes at the external surface have previously been observed after reductive activation[29]. However, due to the ex situ nature of earlier studies, it remained unclear whether these changes occurred during catalyst activation, ammonia synthesis, reaction quenching, or were artifacts resulting from sample transfer. Moreover, it was not evident whether the compositional evolution was confined to the surface or extended into the catalytic bulk[29]. Our in situ study demonstrates that these changes occur early during the activation and affect the complete catalytic structure, although the most evident segregation of promoters is found at the external material.

We hypothesize from the TEM images (Fig. 5j, k) that part of the promoter phases remained also inside of the metallic phases, producing lattice distortions and strain. This effect has important

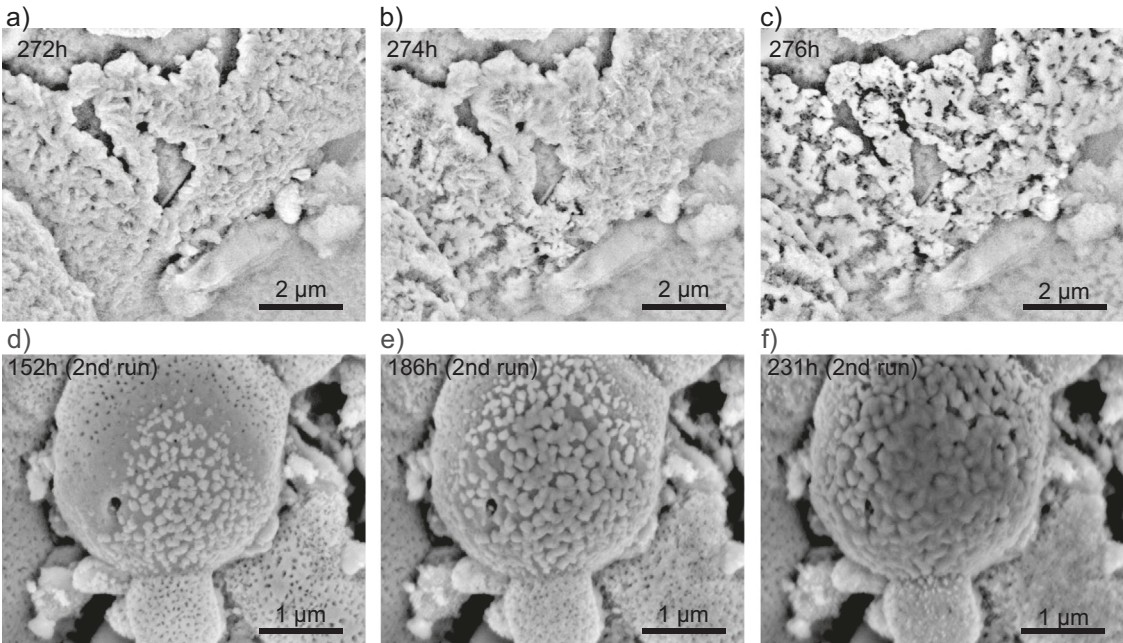

**Fig. 4 | Continuation of the in situ SEM observation of the catalyst surface at iron-rich granules under reaction conditions. a–c** Morphological transformation of the segregated material after TOS = 272 h. The images in (**a–c**) were extracted from Supplementary Movie 4. **d–f** Catalyst surface imaging during isothermal treatment at 500 °C in a second experiment shows phase exsolution and platelet segregation. Images in (**d–f**) were extracted from Supplementary Movie 5. Conditions of acquisition: 22 Pa, $H_2:N_2:Ar = 3:1:0.1$.

consequences on the final nanostructure and the catalytic potential of the active surface. For instance, the strained material can lead to surface steps (Fig. 5k) on which the reaction is faster due to the prevalence of active $C_7$ sites[12,45–49].

Our overview of the cross-section (Fig. 5a–h) shows homogeneous elemental ratios of Al, Si, and Ca along the catalyst granule. Furthermore, the changes of the NAP-XPS spectra of these elements (Fig. 6d–f and 7) and the disappearance of aluminum hydroxides in favor of calcium (silico)aluminates (Supplementary Fig. 2) confirm the formation of phases combining the three promoter elements in the final catalyst[31]. The role of these mineral phases relies on preventing agglomeration/sintering of the iron crystallites while preserving the 3D nanodispersion of the porosity (Fig. 5j, k). Hence, they also exert the templating action on the active structure and are identified here as the main reason of the structural longevity of the technical catalyst.

The mineral phases have compositions similar to known cementitious materials, which can explain the observed morphological evolution of the external surface detected in our study (Supplementary Movies 1–5 and Figs. 1d, 3–5). The water produced by the reduction could react with these phases in situ, giving rise to a sequence of hydration, precipitation and cement setting reactions[50,51]. It is known that cement hydration induces several structural morphologies depending on the conditions, mixture composition, and aging time[38,51–53]. For instance, the micrometric morphologies produced during hydration and aging of a typical cement sample (CaO, $SiO_2$, $Al_2O_3$, and $Fe_2O_3$) can be described by foil/flake/platelet structures at the initial stages of hydration, and interlocked foils, disordered/amorphous material, and needle-like structures with increasing aging time[50]. It has been shown that nanofiber and nanosheet polymer structures of the hydrated product are the basic units of the cement phase, which can subsequently assemble to give rise to platelets and needle-like materials. Despite the highly divergent treatment conditions, we found strikingly similar morphologies during the catalytic activation, including an initial phase segregation followed by the platelet morphologies, and finally a mixture of interlocked platelets, amorphous material, and needle-like structures (Vid. 3, 4, Figs. 3–5).

These observations are compatible with ongoing hydration, setting, dehydration and restructuring of cementitious phases during the treatment, as represented schematically in Fig. 8a.

The hydration of the mineral phase is initiated by the water stemming from the $FeO_x$ reduction. The early stages of hydration eventually give rise to the assembly of the platelets. The internal arrangement of the platelet shows a stacking of fiber-like units (Fig. 5i) that may correspond to the nanofibers of the cement setting. Upon further hydration, cross-linking occurs. At much longer TOS when the water concentration becomes sufficiently small due to the completion of the $FeO_x$ reduction, the platelet integrity is affected by dehydration, giving rise to the disordered crust-like material characteristic of the spent catalyst (Figs. 1d, 4a–c, and 8a). A hint to the platelet disintegration can be deduced from the partly amorphized layer covering the platelet before the collapse of the structure, indicating the initiation of dehydration (see amorphous rim in Fig. 5i). The setting of the resulting disordered crust-like phases (Fig. 4a–c) did not bring any detectable changes in the catalytic conversion (blue region in Fig. 2), suggesting that the production of ammonia was not related to the external material, but rather to the porous metallic substrate. However, the identified promoter phases are highly effective drying agents[29,54], contributing to the chemical resilience of the catalyst and allowing it to withstand the moisture levels typically encountered in industrial processes[29].

Chemically, these processes at the material enriched by the promoters imply transformation of short calcium silicate/silicoaluminate units into polymeric materials, in line with the trends of XPS data (Fig. 6). The presence of alkaline phases such as CaO, $Ca(OH)_2$, and KOH can mitigate the self-poisoning of the reaction by tuning the surface affinity against ammonia adsorption. Interestingly, the potassium phase was found to transform from $K_2O$ into a mixture of KOH and "ammonia K" (Fig. 6c). The formation of KOH can be explained by the facile hydration of $K_2O$ even at room temperature. However, "ammonia K" was only detected under reaction conditions. Based on its binding energy in the XPS spectra, the observed species could be attributed to metallic K. However, under the conditions of ammonia

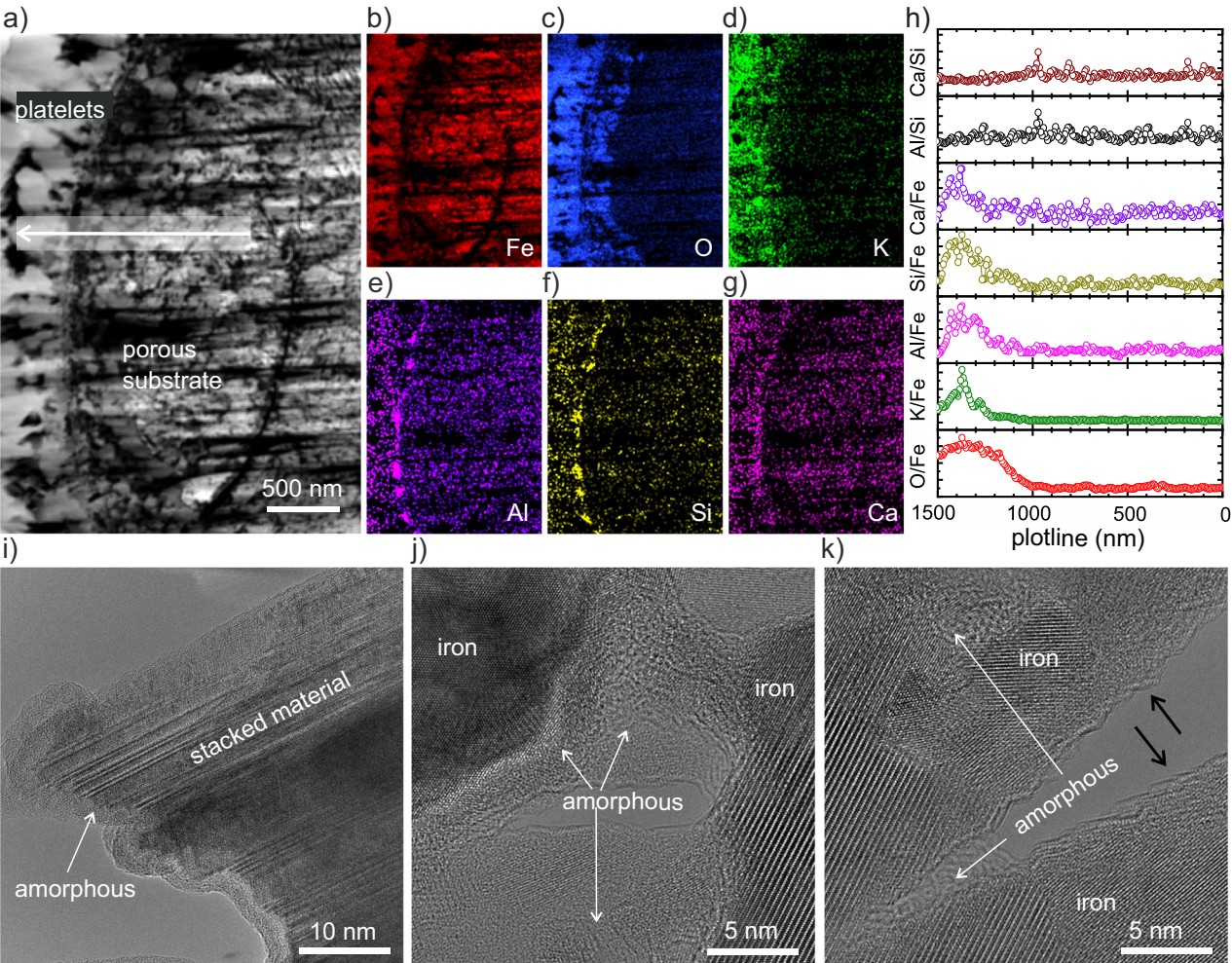

**Fig. 5 | Structural and compositional investigation of a thinned cross-section of an iron-rich granule of the spent catalyst after the OSEM experiment. a** A high-angle annular dark field scanning transmission electron microscopy (HAADF-STEM) image of the cross-section. The arrow marks the direction of the plotline used for the analyses of the energy dispersive X-ray spectroscopy (EDX) measurements presented in (**h**). **b**–**g** EDX elemental maps corresponding to Fe, O, K, Al, Si, and Ca, respectively. **h** Plot of elemental ratios for O/Fe, Al/Fe, Si/Fe, K/Fe, Ca/Fe, Al/Si, and Ca/Si calculated by integration of EDX intensities of (**b**–**g**) along the plotline represented in (**a**). **i** A high resolution TEM (HR-TEM) image of a platelet particle showing an ordered structure (stacked nanofibers) surrounded by an amorphous layer. **j**, **k** HR-TEM images of the porous catalytic substrate revealing the presence of metallic iron crystallites held together in a porous arrangement by an amorphous material. The black arrows in (**k**) indicate surface steps of ammonia iron.

synthesis—where water and additional oxidic phases are present—the presence of metallic K is unlikely[16,25,55,56]. In one study[55], metallic K was deposited onto an Fe single crystal catalyst, resulting in a net increase in the electronic density of the iron. This enhancement was proposed to improve $N_2$ dissociation, explaining the higher activity of the K-promoted catalyst as compared to pure iron, and the higher concentration of dissociated N found on the K-promoted iron single crystals[57]. Although metallic K exhibited a strong promoting effect, the former studies concluded that its formation from oxidic precursors under realistic reaction conditions was unlikely. In another study[58], metallic K was found to have a stronger promotion effect than $K_2O$, but it was found to be volatilizing during heating and was lost from the surface. Similarly, elemental potassium was reported to be unstable on the iron surface, volatilizing during thermal treatment[59]. However, coadsorption of oxygen was shown to stabilize sub-monolayer coverages (0.10−0.30) of potassium, regardless of surface orientation. Additional single-crystal studies[56,60] confirmed that adsorbed metallic K significantly enhanced the rate of nitrogen adsorption but desorbed under stationary reaction conditions above ~400 °C. Addition of oxygen to iron pre-covered with potassium and also addition of potassium to iron pre-covered with oxygen were found to thermally stabilize the adlayer. A more recent study using supported iron nanoparticles mixed with alkaline oxide promoters ($K_2O$, $Li_2O$, $Na_2O$, $Cs_2O$) indicated that the promoters formed surface patches on the iron nanoparticles of the spent catalyst[61]. DFT simulations revealed that a stabilizing Fe−O−K interaction was required, leading to a double-layer structure with oxygen atoms forming the first sublayer and $K^+$ ions positioned externally. In the same line, an XPS study of $K^+$ ions on a mica substrate reported similar spectral features to those observed here[35]. The authors discussed that the signal in the position of "ammonia K" may correspond to undercoordinated cations located at the surface.

Some studies have also proposed that the potassium phase may transform into potassium amide ($KNH_2$) through a two-step process involving the initial formation of metallic potassium, followed by its reaction with ammonia to form the amide[58]. This hypothesis was examined through detailed equilibrium measurements under varying potassium promoter loadings[62]. Additionally, $KNH_2$ was introduced directly into the reactor to assess its influence on ammonia synthesis. The study concluded that neither $KNH_2$ nor metallic K were formed under any of the investigated conditions[62].

Based on these arguments, we propose that "ammonia K" corresponds to a highly dispersed form of a KOH adsorbate residing on the

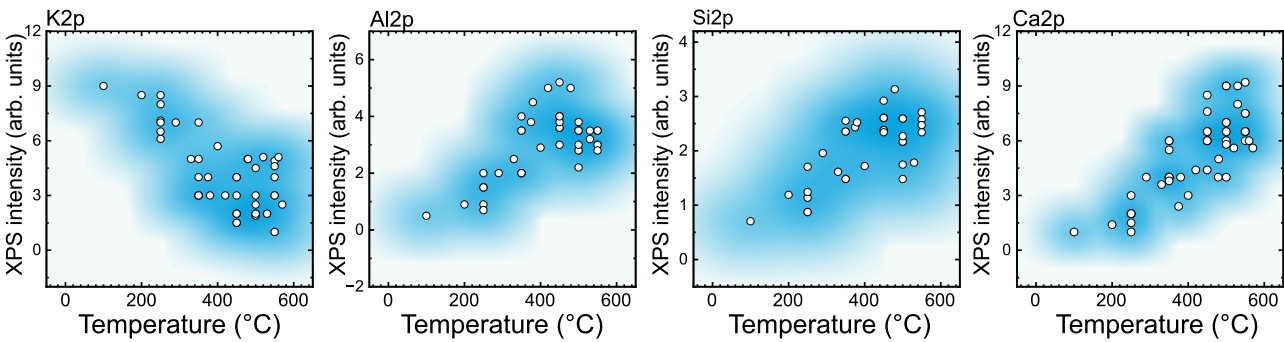

**Fig. 6 | Variation of chemical composition of the catalyst surface under reaction conditions.** NAP-XPS spectra of **a** Fe2p, **b** N1s, **c** K2p, **d** Al2p, **e** Si2p, and **f** Ca2p. From bottom to top, spectra display data collected at 250 °C, 500 °C, and after cooling to room temperature, respectively. Conditions of acquisition: 50 Pa, Deuterium:N₂:Kr = 3:1:0.1. Further details are given in Supplementary Table 2.

**Fig. 7 | A compilation of the integrated peak intensities corresponding to K, Al, Si, and Ca as a function of temperature as obtained from NAP-XPS measurements.** Conditions of acquisition: 50 Pa, H₂:N₂:Kr = 3:1:0.1. The data dispersion was given a contour map (95% density) for visualization.

Fe surface under reaction conditions. This surface species forms in situ during activation due to hydration of K₂O. During activation the moisture level in the feed can reach values around $10^2$ ppm[29], supporting the prevalence of KOH. The melting point of KOH ($T_m = 406\,°C$) suggests that at some point of the treatment it may become liquid or highly mobile, giving rise to the characteristic spectral feature observed in the XPS (Fig. 6c). Hence, with increasing temperature, KOH may migrate into the substrate of the catalyst, where it is adsorbed on the iron. Consequently, the XPS measurement senses a decrease in K contents at the external surface (Figs. 7 and 8b). At decreasing temperatures, the KOH can be segregated back onto the oxidic material and on the external surface (Figs. 7 and 8b). This high mobility of KOH reconciles the obvious K segregation at the external surface found in the present study as well as in previous investigations of spent catalysts[24] with its apparent vanishing from the external surface under working conditions (Figs. 7 and 8b).

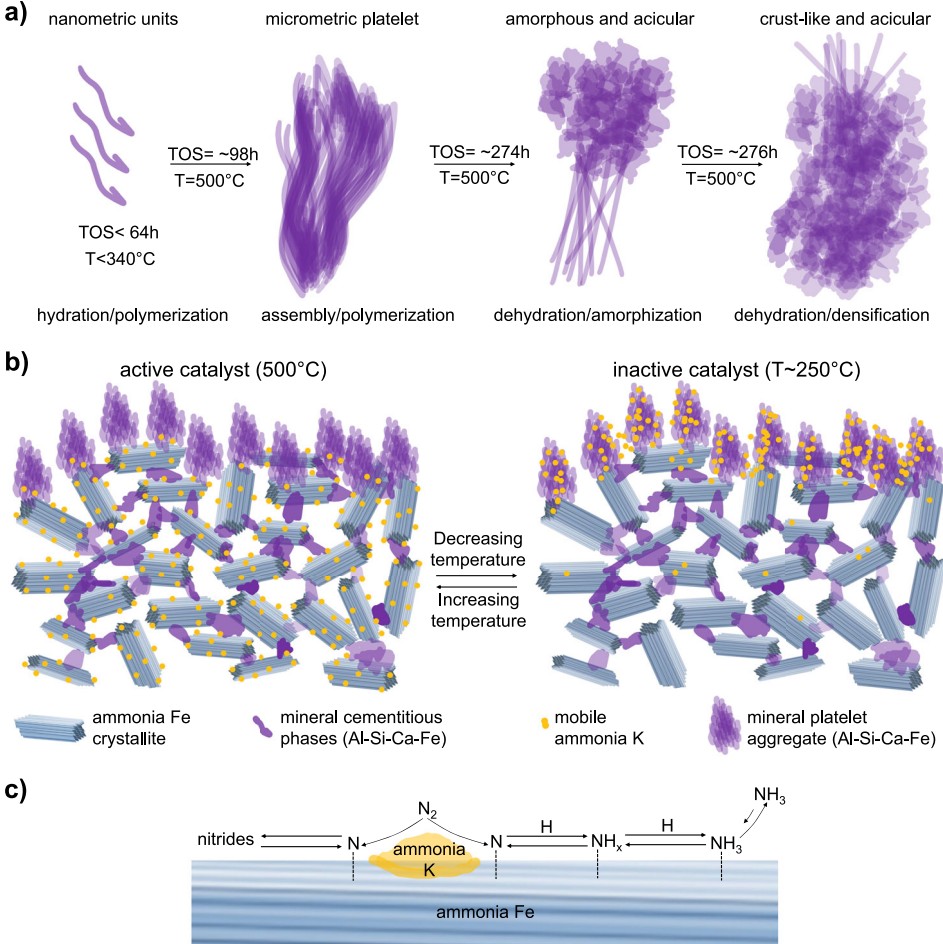

**Fig. 8 | Schematic representations of the morphological evolution of the promoter phases, the activated catalyst architecture, and the catalytic production of ammonia. a** The chemical linking of Al, Si, Ca, and Fe oxidic phases produces a material similar to a cementitious phase. At early stages of hydration, this phase gives rise to nanofiber basic units. With ongoing aging (TOS) under decreasing water vapor the basic units assemble sequentially in platelets, needle-like aggregates, and densified/disordered material. **b** The activated catalyst exhibits a 3D nanodispersion of ammonia Fe crystallites structurally stabilized by a set of cementitious phases. Excess of the segregated promoter material accumulates at the external surface in the several forms depicted in (**a**). Ammonia K covers the ammonia Fe of the catalytic substrate at reaction conditions (500 °C), but preferentially accumulates on the external surface at low temperatures. **c** The dissociative chemisorption of $N_2$ is enhanced by in situ formed ammonia K/ammonia Fe phases due to electronic transfer effects. The atomic N species reacts either sequentially with atomic H to give rise to the ammonia product, or with the Fe surface to produce nitrides. The presence of alkaline KOH increases the desorption of the ammonia product.

Therefore, it is now possible to track the promoter effects on ammonia synthesis. It is known that the rate limiting step is the nitrogen dissociation, which is highly structure sensitive on iron single crystals[11,63,64], and gives rise to catalytic turnover of ammonia production in the ratio 418:25:1 for the Fe(111), Fe(100), and Fe(110) surface terminations, respectively[28,65]. Notably, the presence of potassium on Fe(111) and Fe(100) increases the rate of ammonia synthesis[25], and reduces the surface orientation sensitivity of the reaction. A highly mobile KOH adsorbate promoting the active site can explain this behavior, because it can migrate freely on the Fe surface at ammonia synthesis conditions, regardless of the surface orientation. This would level-out the structure sensitivity of the reaction.

It also seems possible that the beneficial effect of K relies on improving the rate of nitrogen dissociation[57]. The KOH adsorbate can produce a modification on the nature and number of active sites, for instance, by increasing the local electron density of the iron surface and decreasing its work function[25,26,49,55,59]. Activation of $N_2$ involves weakening its triple bond[7], for which electron-rich mediators such as alkaline hydroxides, electrides, hydrides[66,67], etc., are beneficial. The mechanism of this effect may be either by direct electronic transfer into antibonding orbitals of $N_2$[7], or by an electron transfer into the iron

surface that subsequently destabilizes the triple bond by π-backdonation[1,11,55]. The modification of the active surface can also induce higher rates of $NH_3$ desorption, thus preventing the self-poisoning observed at high output conditions. For instance, in situ studies using infrared spectroscopy on Fe single crystal catalysts covered with preadsorbed K found a measurable redshift in the ammonia-related frequencies[68,69], indicative of a weakening in the ammonia iron bond. The authors concluded that the presence of K was increasing the desorption rate of ammonia and the rate of hydrogenation from N into $NH_x$. Similarly, a theoretical study on alkaline metal promotion found that dipoles could be created on the active surface during the electronic transfers of the reaction[70]. These dipoles electrostatically attract the transition state of dissociating $N_2$, while simultaneously exert a repulsion on the $NH_x$ fragments, leading to enhanced product desorption[26]. Another possibility is the local geometric influence of the adsorbates on the iron surface, favoring the occurrence of facets and high index planes that are more active to ammonia synthesis[71]. A combination of these influences can explain the increased catalytic performance.

The KOH adsorbate can also explain the fact that ammonia iron is almost indistinguishable from α-Fe at the unit cell and bulk

structures[10,17], since the adsorbate only affects the topmost layers of the iron phase without inducing important changes in the general structure of the crystal lattice. This is stressed out by the new finding that the active phase of the technical catalyst only exists in situ.

While the KOH/Fe structure enhances the turnover of nitrogen dissociation, the atomic N generated by this influence can cascade rapidly into the $NH_3$ product by sequential hydrogenation (Fig. 8c). We detected atomic N, $NH_x$ fragments and nitrides under reaction conditions (Fig. 6b). Although it has long been debated that iron nitrides could be key reaction intermediates[2], our characterization demonstrates that the dominant N species under synthesis conditions was the atomic N. The nitride phases became dominant only at low temperatures. This observation might point to a parallel path involving the nitrides that is unfavorable for ammonia synthesis (Fig. 8c). These nitrides, however, can be converted to atomic N and $NH_x$ by thermal activation in presence of H, thus explaining the ammonia production on samples of $FeN_x$ treated in pure $H_2$ feed[2,33,72]. This further aligns with the rapid decrease of nitride species during the activation procedure induced by the thermal cycling (Supplementary Fig. 7b).

Our interpretation does not imply that the nitrides are simply an unavoidable consequence of the chemical potential. We think that they are key at the early stages of activation for preventing the rusting of activated metallic iron by the evolving water produced by the reductive treatment. These nitrides may also affect the catalytic rate of ammonia formation, for instance, by initiating the catalytic activation, or by tuning the surface into a state active for hydrogenation. Hence, when the reaction leaks into the formation of nitrides, the resulting effect could reactivate the ammonia path, thus decreasing the propensity of further nitride accumulation. A hint to this dynamic was recently presented[12], demonstrating that the active sites continually appear and disappear under the effect of the local chemical potential. This transient existence of the dissociation site prevents the accumulation of stable nitrides that poison the catalyst. Simultaneously, nitrogen atoms could be diffusing into the subsurface of iron at the high temperatures of the process. Hence, the N states on Fe can be better described by a kinetic loop similar to a frustrated phase transition[21,73] involving fluctuating compositions of nitrides, subsurface species, and N+$NH_x$ surface species. This behavior may explain both the long-term stability of the working catalyst and its pronounced sensitivity to poisoning. We believe that similar dynamics are also the reason to the broader versatility of Fe−N systems, which have already been harnessed in other applications, such as battery materials[74].

Finally, we mention that the activity of the multi-promoted industrial catalyst in the low-pressure OSEM experiments (Supplementary Table S1) approaches the levels reported for single-crystal iron catalysts at 20 bar, although it remains lower. However, drawing conclusions about the intrinsic activity of specific catalytic sites in any of these systems is very challenging. Variations in activity may arise from differences in the active site density or intrinsic site reactivity. This is the case not only in the as-prepared state of the precatalyst material, but also in its working state, were an evolution of the structure takes place. The comparison is even more complicated when considering the multi-promoted catalyst due to the influences of the additives that affect both, the texture and the electronic properties. Because the active sites are transient and form only in situ[12], in situ characterizations sensitive to the fast turnover of the sites are essential for determining the intrinsic activity. Additionally, the reaction rate in ammonia synthesis is influenced by many external factors, making a comparison of results across different conditions unreliable. Despite this, the qualitative agreement between our OSEM and high-pressure data acquired on Fe single-crystals is encouraging. The present study emphasizes the need of further developing operando characterization techniques compatible with increasingly higher pressures in order to bridge the pressure gap.

In summary, our study demonstrates that the activation process is the truly critical step where the catalyst is formed by simultaneous interplaying dynamics. It is the confluence of all these elements that only together form this unique catalyst. These elements comprise a phase segregation exerting a templating action favorable to open porosity, a controlled kinetics of reduction for homogeneous iron dispersion, formation and setting of cementitious phases for structural integrity and durability, iron nitridation at the beginning of activation for rusting prevention, and highly mobile alkaline species for enhanced activity. Our operando characterization has revealed these hidden aspects, providing a novel dimension of insight, with the consequence of valuing the details of synthesis and activation procedures for the maximum attainable performance in prospective researches.

The summary picture derived from the present work may be used to optimize the creation of active sites, for instance, by delicate tuning the content and distribution of the promoters for an optimized activation procedure, and a correct kinetics of the cement-forming phase that may be reduced in its abundance without damaging the nanostructural integrity. Another approach may include a bottom-up construction of the catalytic entity from layered iron nanosheets structurally stabilized by the correct type, load, and location of the promoter phases.

Our investigation bridges the material gap and constitutes an effort towards bridging the pressure gap in heterogeneous catalysis. In fact, our study highlights that ammonia iron is not simply a particular arrangement of the α-Fe phase, but a dynamic entity that only exists in situ by the synergistic action of the various components of the technical, multi-promoted catalyst. We anticipate that further advancements in operando techniques will ultimately close the pressure gap in the coming years. Our study further shows the relevance of investigating complex technical formulations to establish more realistic structure-function correlations of the catalytic phenomenon.

## Methods

### Preparation of the initial catalyst
The multi-promoted catalyst precursor was synthesized in an electric arc furnace using industrially relevant formulations and scale. After arc melting the fine powder mixture, the melt was quenched[27]. The fresh solid catalyst was cut into aliquots ($8 \times 6 \times 1\,mm^3$) with a diamond saw, and the aliquots were treated in an atmosphere of $H_2$ at 1 bar for 5 h at 500 °C. Afterwards, the sample temperature was reduced to 60 °C and the catalyst was passivated in a stream of 1% $O_2$ diluted in Ar for safe transport. The passivated aliquots were used as initial catalysts for OSEM and NAP-XPS experiments.

### Operando SEM set-up
An aliquot of the initial technical catalyst was positioned inside a customized flow reactor compatible with SEM imaging. Our set-up features a quartz tube reactor inside the chamber of a commercially available environmental SEM (ESEM, FEI 200 Quanta FEG) lined to a quadrupole mass spectrometer (QMS, 200 Prisma Pfeiffer). The catalyst was heated by illumination with an infrared laser (808 nm, maximum 110 W)[75]. Changes at the catalyst and in gas phase composition are determined simultaneously, enabling the direct investigation of the influence of catalytic dynamics on performance.

We used a metallic stainless-steel mesh to support the catalyst aliquot inside of the tube, and spot-welded a K-type thermocouple at the catalyst position for temperature detection. Pure gases (Westfalen 5.0) were dosed into the quartz tube reactor by individual mass flow controllers (Bronkhorst), and the gas lines were equipped with $O_2$/moisture filters (Agilent in-line purifier OT3-2) to ensure gas quality.

Images of the catalyst surface were acquired with the large field detector of the ESEM every 17s−68 s at acceleration voltages of 7.5−10 kV at a chamber pressure of 22 Pa and a pixel depth of 8 bits.

For the operando experiments, the aliquot of the passivated technical catalyst (179.1 mg and 142.4 mg after reduction) was treated in an atmosphere composed of 3.0 mlN min$^{-1}$ H$_2$, 1.0 mlN min$^{-1}$ H$_2$, and 0.2 mlN min$^{-1}$ Ar. The gas flows were dosed by mass flow controllers into the reactor until the QMS signals had stabilized. The catalyst was heated to 500 °C at 12 K h$^{-1}$ while being imaged (TOS = 34 h). Due to thermal expansion effects, the field of view was found to be continually drifting and the images went into defocused conditions during the temperature ramping. At the end of the heating ramp, the catalyst was treated isothermally during 225.2 h under continual imaging in the ESEM. Two experiments were performed.

After the second experiment, the reactor temperature was decreased at 10 K min$^{-1}$ under the reaction feed. The spent sample was recovered from the reactor in a glass vial previously purged with Ar, and the catalyst was stored in a glovebox with Ar atmosphere for subsequent characterization.

The collected video frames were treated with Fiji[76]. The stack of video frames was aligned with translation registration. Afterwards, the image noise was reduced with a bandpass Fourier filter (min: 2, max: 56). The treated stack was exported in video file format.

### Near-ambient pressure X-ray photoelectron spectroscopy

NAP-XPS measurements were performed at the UE56/2-PGM1 (Elliptical Undulator) beamline of the synchrotron radiation facility of BESSY II of Helmholtz-Zentrum Berlin, Germany. The details of the beamline layout and performance can be found elsewhere[22]. The home-built near-ambient pressure electron spectrometer is described in detail in the literature[23].

Our NAP-XPS setup has a spatial resolution (150 μm × 80 μm) similar to the field of view of Fig. 1a, and operates under reaction conditions comparable to the OSEM reactor[21]. Due to the time restriction in synchrotron experiments, the NAP-XPS characterization covers only the earlier stages of the activation procedure, extending in total to TOS = 126.5 h (see Fig. 2 for comparison).

An aliquot of the initial catalyst and a K-type thermocouple were fixed into a sapphire sample holder. The sample was mounted inside the XPS/XANES reaction cell, near to the aperture of the first differential pumping stage. The exit slit of the beamline was 180 μm and a pass energy of 10 eV and 0.1 eV step were used during spectra acquisition with an experimental resolution of 0.55 eV. The heating treatment was carried out with an infrared laser (808 nm, maximum 60 W) shining at the rear of the mounting plate.

The catalyst was treated in a reaction mixture composed of Deuterium 3.0 mlN min$^{-1}$: N$_2$ 1.0 mlN min$^{-1}$: Kr = 0.1 mlN min$^{-1}$ (controlled by Bronkhorst mass flow controllers) using a heating rate of 0.5 K min$^{-1}$ to a temperature of 550 °C. Afterwards, the catalyst was isothermally treated at different temperatures in the following order: 500, 450, 350, 250, 550, 450, 350, and 500 °C. Experiments on different aliquots of the same sample were also performed at temperatures of 100, 380, 420, and 470 °C.

XP spectra of core level regions were recorded with kinetic energies of emitted photoelectrons at 150 eV and 300 eV for Fe 2p, N 1s, K 2p, Al 2p, Ca 2p, and Si 2p. The spectra of N 1s were acquired after accumulation of 200 scans.

The binding energies were calibrated to the Fermi edge of an Au reference sample. The spectra were deconvoluted with combined Gaussian and Lorentzian functions after a Shirley + linear background subtraction and the peak positions were fitted from known values in the literature using the Fitt software package.

### Cross-section manufacture and transmission electron microscopy investigation

For TEM investigations, a lamella of the spent catalyst from the second OSEM experiment was prepared with an FEI Helios NanoLab G3 FIB/SEM system using Ga ions with energies up to 30 keV. A Pt-C protective layer of 600 nm thickness was added to the ROI by electron beam induced deposition. Additional 600 nm of carbon were deposited by FIB to ensure sufficient protection during the subsequent ion milling. The lamella was thinned from both sides to a thickness below 100 nm by a 30 kV Ga ion beam. In the final cleaning, low-energy ion beams of 5 kV and 2 kV were applied to the sidewalls of the lamella. The prepared lamella was examined with a double aberration-corrected JEOL JEM-ARM 200CF TEM operated at 200 kV with an emission current of 10.0 μA, and with a Talos F200E TEM at 200 kV equipped with STEM and electron X-ray energy dispersive spectroscopy (EDX) detection. High-angle annular dark field (HAADF) STEM images were acquired with a semi-convergence angle of 20 mrad and collection angles of 80 mrad (inner) and 320 mrad (outer).

## Data availability

The data generated in this study have been deposited in the AC/CATLAB Archive database and can be downloaded from https://ac.archive.fhi.mpg.de/D64655.

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

## Acknowledgements

We are grateful to the Federal Ministry of Education and Research, in the framework of the project Catlab (03EW0015B) and to the Deutsche Forschungsgemeinschaft (DFG, German Research Foundation) under Germany's Excellence Strategy–EXC 2089/1–390776260 for financial support. We also acknowledge Helmholtz-Zentrum Berlin for providing beamtime in the frame of proposal ID 191-08444, and for the continuous support of the (near) ambient pressure XPS activities of the MPG at the BelChem-PGM beamline.of the synchrotron light source BESSY II, Germany.

## Author contributions

L.S.D. performed the operando SEM experiments. R.B., K.D., and L.S.D. carried out the NAP-XPS experiments. J.F. performed the ex situ activity measurements and the thermodynamic calculations. Z.G. and D.I. prepared the catalyst aliquots for operando SEM and NAP-XPS measurements and the ex situ SEM characterization. A.H. prepared the thin cross-section. M.B. and K.D. did the TEM imaging. F.G. performed the XRD analysis and the phase assignment. R.E., S.R., and A.R. synthetized the industrial catalytic precursor. L.S.D., R.B., F.G., A.K.-G., R.S., and T.L. analyzed the operando spectromicroscopy and post-reaction data. R.E., S.R., A.R., R.S., B.R.-C., H.R., A.K.-G., and T.L. guided the research. L.S.D. and T.L. wrote the manuscript. All authors discussed the results and commented on the manuscript.

## Funding

## Competing interests

The authors declare no competing interests.
