## [Transparent Peer Review file · Nature Communications]

Decoding Technical Multi-Promoted Ammonia Synthesis Catalysts

Corresponding Author: Dr Thomas Lunkenbein

Version 0:

Reviewer comments:

Reviewer #1

(Remarks to the Author)

The manuscript used in-situ characterization techniques to deeply reveal the structural evolution of ammonia synthesis catalysts and analyze the distribution and role of different promoters during the activation process. I recommend an acceptance of this manuscript for publication while several points need to be addressed.

1. The ICP results of catalyst evolution at different temperatures should be provided to more accurately indicate the pattern of element changes.
2. To more accurately demonstrate the overall changes in catalyst particles and the distribution of product phases after the reaction, more experimental evidence such as CT, FIB, etc. is needed.
3. The specific mechanism of the interaction between the "ammonia K" on the catalyst surface and the iron catalyst is not yet clear. Suggest studying changes in the electronic properties of sites through in-situ infrared spectroscopy (IR) or Raman spectroscopy.
4. Figure 4h shows the changes in element distribution of the spent catalyst, but does not explain the potential relationship between these changes and ammonia production and long-term stability. Please analyze the correlation between the structural characterization results of the spent catalyst and its operational performance (such as conversion rate and selectivity).
5. The synergistic effect between the distribution of nano-dispersed iron and local potassium enrichment was mentioned, but there is a lack of foundation for quantifying the catalytic conversion rate and catalytic site density. Suggest combining experimental calculations to provide measurement data on the density of surface points per unit area, to enhance the persuasiveness of the discussion.
6. Is the Fe-N interaction only related to the type and distance of atoms, and how is it different from ion intercalation and adsorption in batteries (e.g. Energy Environ. Mater. 2023, 6, e12342. <https://doi.org/10.1002/eem2.12342>; Adv. Mater. 2021, 33, 2100171. <https://doi.org/10.1002/adma.202100171>)? When Fe exists in the form of ions on the surface of the catalyst, and the distance between it and N is close, what impact does it have on the intermediate adsorption products and catalytic yield of Fe-N? By combining the aforementioned literature, the authors are encouraged to make a brief discussion with an enhanced multidisciplinary impact of their work.

Reviewer #2

(Remarks to the Author)

This manuscript presents a significant advancement in the understanding of industrial ammonia synthesis catalysts, particularly multi-promoted iron-based systems. Through operando scanning electron microscopy and near-ambient pressure X-ray photoelectron spectroscopy, the authors provide good insights into the structural evolution and dynamic behavior of these Fe-based catalysts under reaction conditions. I recommend it for publication with minor revisions.

Strengths: (1) The activation process is essential for forming the active catalyst configuration. The active phase consists of a nanodispersion of metallic iron covered by mobile potassium-containing adsorbates termed "ammonia K". (2) In situ studies reveal that potassium species exhibit high mobility, redistributing between the catalyst surface and the bulk during reaction.

Suggestions: (1) For the same question, the author has explored multiple possible explanations. It is recommended that they ultimately present the interpretation they think most compelling. "For K species (Fig. 5c), the 2p_{3/2} contributions indicate a mixture of mainly K₂O (BE= 293.2 273 eV) and KOH (BE=293.8 eV) at 250°C. At 500°C, emergence of a peak at BE=294.75

eV was observed. This signal, which we refer to as “ammonia K”, may indicate the formation of metallic K [30] or undercoordinated isolated surface K⁺ species [31].” “Previous studies have suggested that K at reaction conditions should exist as an anhydrous KOH adsorbate [16, 25,49] [50], because the metallic form would be volatilized.” There is a potential contradiction in the interpretation of “ammonia K.” Is “ammonia K” metallic K or KOH or a coordinated complex? At the end of the manuscript, the author used KOH to explain the experimental phenomenon. Could “ammonia K” be KNH₂ or K₂NH? If metallic K existed in a free state, it would likely evaporate under reaction conditions (400-500°C). However, if K is strongly interacting with the surface (Fe or oxides), it might exhibit a higher stability and remain localized. (2) “It also seems possible that the beneficial effect of K relies on improving the rate of nitrogen dissociation. The KOH adsorbate can produce a modification on the nature and number of active sites, for instance, by increasing the local electron density of the iron surface.” Is there any experimental basis for this statement? Please cite relevant reports. (3) “Our investigation is an effort to bridge the pressure and material gaps of heterogeneous catalysis, confirming that ammonia iron is not simply a particular arrangement of the α -Fe phase, but a dynamic entity that only exists in situ by the synergistic action of the various components of the technical, multi-promoted catalyst.” The authors claim to bridge the pressure gap in heterogeneous catalysis; however, their operando experiments were conducted at only 22 Pa, which is far from industrial ammonia synthesis conditions (200-300 bar). I suggest the authors revise the above sentence to more accurately reflect the experimental conditions. (4) About Fig.7b, if K accumulates on the surface of mineral platelet aggregate (Al-Si-Ca-Fe) rather than on the ammonia Fe surface, how does K enhance ammonia synthesis performance?

Version 1:

Reviewer comments:

Reviewer #1

(Remarks to the Author)

The authors have made the necessary revisions, and I am satisfied with the revised version and reply. I recommend its publication in this form.

Reviewer #2

(Remarks to the Author)

I commend the authors for their thoughtful and thorough revisions. All my concerns have been addressed, and I find the current version suitable for publication without further changes.

Response to comments on manuscript "Decoding Technical Multi-Promoted Ammonia Synthesis Catalysts"

All remarks will be discussed below in a point-to-point response. Specific responses to the reviewers' comments are emphasized by blue text and changes in the manuscript are highlighted in yellow.

Point-point answer to the reviewers' comments:

Reviewer #1 (Remarks to the Author):

The manuscript used in-situ characterization techniques to deeply reveal the structural evolution of ammonia synthesis catalysts and analyze the distribution and role of different promoters during the activation process. I recommend an acceptance of this manuscript for publication while several points need to be addressed.

1. The ICP results of catalyst evolution at different temperatures should be provided to more accurately indicate the pattern of element changes.

We thank the reviewer for the time employed assessing our manuscript and for the constructive feedback that helped to improve its quality. ICP is a very important technique as it delivers quantitative information about the composition of a sample. ICP can be used in combination with a mass spectrometer (ICP-MS) or with an optical light spectrometer (ICP-OS). However, in both cases, the sample has to be dissolved in a liquid. Hence, its structure is destroyed. Thus, we consider that this method would not serve to answer the questions about the phase segregation discussed in our study.

Alternatively, ICP can be used as an excitation for etching solid samples, i.e., ICP Reactive Ion Etching or ICP-RIE (see: K. Racka-Szmidt, B. Stonio, J. Żelazko, M. Filipiak, M. Sochacki, A Review: Inductively Coupled Plasma Reactive Ion Etching of Silicon Carbide, in: *Materials*, 2022.). Here, the etching takes place by localized vaporization of a selected region and the material underneath is exposed. The results that can potentially be achieved by this approach are of a similar nature to those achieved with the Focused Ion Beam (FIB) milling approach already presented in Figures 4 and S6 of the manuscript. We carved with the FIB through the catalyst to investigate the elemental compositions and the internal porous structure. The FIB also delivers SEM images of the material while the cross section is being prepared. This avoids additional sample manipulation, which can damage its structure. We mention that the observed changes of the phase segregation were found *during* the reaction. An *operando* ICP approach, where the ICP method is integrated while the reaction is taking place, is highly desirable but it is yet to be developed.

Regarding the question about the elemental segregation of the promoter phases, we must mention that our observations agree with results discussed in previous studies such as Ref. 29, & 48. It has been observed that the atomic surface composition changed after activating the catalyst with H₂ or with H₂+N₂ mixture. Furthermore, the

surface compositional variations depended on the final treatment temperature. Hence, the piece of information missing in these experiments was not the elemental segregation, but understanding how and when these changes occurred, i.e., if they occurred during the catalyst activation, during the ammonia synthesis, during the reaction quenching, and if these compositions were only related to the surface or if they extended also to the catalytic bulk (Ref. 29). For these reasons, we opted to address the catalyst evolution with the presented combination of NAP-XPS and OSEM, delivering a correlative in situ characterization of the catalyst transformation under the reactive stimulus. Our data is further confirmed by our TEM analysis of the cross section in the spent catalyst material.

Accordingly, we added the following sentences to the manuscript text:

line 339

Compositional changes at the external surface have previously been observed after reductive activation [29]. However, due to the ex situ nature of earlier studies, it remained unclear whether these changes occurred during catalyst activation, ammonia synthesis, reaction quenching, or were artifacts resulting from sample transfer. Moreover, it was not evident whether the compositional evolution was confined to the surface or extended into the catalytic bulk [29]. Our in situ study demonstrates that these changes occur early during the activation and affect the complete catalytic structure, although the most evident segregation of promoters is found at the external material.

2. To more accurately demonstrate the overall changes in catalyst particles and the distribution of product phases after the reaction, more experimental evidence such as CT, FIB, etc. is needed.

To strengthen this point, we have added catalyst images captured during the preparation of the thin cross section by Focused Ion Beam milling. These images were integrated in Fig. S6 and show clearly the predominance of the platelet aggregates on the external surface, as well as the porous nature of the catalytic substrate.

Fig. S6. a) and b). SEM images of the spent catalytic sample during the preparation and lift-out of a thin cross-section by Focused Ion Beam (FIB) milling. c) and d) BF-TEM overviews of the thinned cross-section showing the lamellar porous structure of the substrate giving rise to open porosity.

It has to be mentioned that the formation of the porous active catalyst is in line with the observed increase of catalyst surface area determined by gas adsorption before and after the activation procedure. Accordingly, we included into the main manuscript the following paragraph:

line 215

Fig. S6a and S6b were acquired during the preparation and lift-out of a thinned cross-section (lamella) of the spent catalyst material from the second OSEM experiment, using Focused Ion Beam (FIB) milling. The images exemplify the segregation of platelet material into the external surface of the catalyst, as observed during the OSEM experiments. Furthermore, Fig. S6b reveals the porous nature of the catalytic substrate.

The formation of an open porous system is consistent with previous gas adsorption measurements, showing a surface area increase from approximately $2\text{m}^2\text{g}^{-1}$ to $15\text{--}19\text{m}^2\text{g}^{-1}$ following reductive activation in multi-promoted systems [16, 29, 30].

3. The specific mechanism of the interaction between the "ammonia K" on the catalyst surface and the iron catalyst is not yet clear. Suggest studying changes in the electronic properties of sites through in-situ infrared spectroscopy (IR) or Raman spectroscopy.

We thank the reviewer for this important suggestion. We agree that the information achievable by IR and Raman spectroscopies is highly desirable to understand specific bond properties from the vibrational characteristics. The suggested techniques (IR, Raman) have already been employed in the determination of the K-promoting effect on iron surfaces. In a study (REF. 67), the authors used in situ reflection absorption infrared spectroscopy (RAIRS) on an Fe single crystal catalyst covered with preadsorbed K. The study found a measurable redshift in the ammonia umbrella mode frequency, indicative of a weakening in the ammonia–iron bond, due to coadsorption with potassium. This result led to conclude that K promotion resulted in an increased desorption rate of ammonia and the rate of hydrogenation from N into NH_x . The same group further concluded that coadsorption of O was needed to stabilize the K, although the specific effect on ammonia desorption dynamics could not be determined (ref. 68). We updated the manuscript accordingly to include these studies:

line 479

For instance, in situ studies using infrared spectroscopy on Fe single crystal catalysts covered with preadsorbed K found a measurable redshift in the ammonia-related frequencies [67, 68], indicative of a weakening in the ammonia–iron bond. The authors concluded that the presence of K was increasing the desorption rate of ammonia and the rate of hydrogenation from N into NH_x .

Please note that in these RAIRS experiments the authors used single-crystal Fe because of its flat geometry. This approach is very challenging to implement with our texturized catalyst geometry found in the reactive environment. Developing these additional operando measurements for the present samples is beyond the scope of this manuscript. Complementarily, another study used in situ Raman spectroscopy to characterize Fe- NH_3 species following ammonia decomposition or synthesis on either pure iron or doubly-promoted $\text{Al}_2\text{O}_3+\text{K}_2\text{O}$ iron catalysts (ref. 71). Interestingly, the authors found Raman bands for Fe-H, Fe- NH_x and Fe- N_2 species in the in situ conditions, although structural bands in the region from $130\text{--}800\text{ cm}^{-1}$ related to Al_2O_3 or K_2O could not be resolved due to strong background artifacts and signal overlapping. This speciation of ammonia fragments (NH_x) on the catalyst surface agrees with our NAP-XPS measurements (Fig. 5).

We add to this discussion that the results found by these prior vibrational spectroscopy approaches are compatible with the results previously found by several studies using surface-sensitive techniques, and that in general these findings can be harmonized with our presented data. In a study (ref. 54), the authors covered an Fe single crystal catalyst with metallic K, finding that a net increase in the electronic density of the iron induced a decreasing work function of the material. Although metallic K exhibited a good promoting effect for N₂ dissociation, the same study concluded that the formation of metallic K adsorbates from the oxidic precursors in the reaction environment of actual ammonia synthesis was unlikely. In another study, the authors found that metallic K was a superior promoter compared to K₂O (ref. 57). Although the catalysts in these experiments were heated during the observations, the diffusion of either K or K₂O into the Fe bulk was not observed. The authors also found that metallic K was lost during the catalyst heating due to volatilization. Following the same line, another study (ref. 58) found that elemental potassium was instable at the iron surface, involving volatilization of the K promoter. However, coadsorption of O was demonstrated to stabilize sub-monolayer amounts (0.10-0.30) of potassium on the iron surface regardless of their orientation. Another study using single crystals (ref. 55) corroborated that adsorbed K accelerated the rate of N₂ adsorption markedly, but would presumably desorb under stationary reaction conditions ($T \geq 670$ K). Addition of oxygen to iron precovered with potassium, and also addition of potassium to iron precovered with oxygen, were found to thermally stabilize the adlayer. A more recent study used supported iron nanoparticles mixed with alkaline oxide promoters K₂O, Li₂O, Na₂O and Cs₂O (ref. 60). The authors found that the promoters were forming patches at the external surface of the iron nanoparticles. Furthermore, their DFT simulations showed in all cases that an Fe-O-K interaction was needed to stabilize the promoter on the surface, adopting a double layer structure where the first sublayer was made of O atoms, while the K⁺ ions remained in the most external layers.

All these results confirm the necessary presence of O species between the K and the Fe. In line with these observations, our results suggest a highly dispersed KOH species in the active state, which we termed as “ammonia K”. Our reasons for choosing KOH as the most likely state are derived from the mentioned literature reports, and the conditions of our experiments, where we detected high mobility of the K species depending on the temperature:

1. In contrast to the model surface science (UHV) experiments commonly found in the prior literature where metallic potassium was directly loaded on the catalyst, we discard the presence of metallic K in our samples during reaction due to its instability and volatility. In fact, any metallic K would immediately react with the product ammonia, giving rise to KNH₂ (ref. 29, 57), which was not observed in our study and has been discarded as a reaction species (ref. 60).
2. K₂O has a melting point of 740°C, which is higher than the reaction temperature of typically 450-500°C. It would be hard to explain the observed mobility of a K₂O phase

while it remains in the solid state. Furthermore, the reduction of the catalyst implies some level of in situ evolving water, which can react with K_2O to produce KOH. Industrially, some hundred ppm levels of moisture are expected in the feed, precluding a “dry” K_2O state in the catalyst. Accordingly, we detected the transformation of K_2O into “ammonia K” in the active state of the reaction, although our experiments were conducted under “dry” conditions, at least after the intensive catalyst reduction.

3. The segregation of K is compatible with the redispersion of a highly mobile species, which has a melting point below the reaction temperature. As discussed in the manuscript text, KOH melts at 406°C , becoming highly mobile at the temperature of ammonia production.

4. There is no evidence of K diffusion into the Fe crystalline bulk as it would be expected for metallic semi-alloying. K diffusion into Fe is negligible even in liquid state at temperatures above 1000°C (J. Sangster, C.W. Bale, B.P. Burton, The Fe-K (Iron-Potassium) System, Journal of Phase Equilibria, 12 (1991) 46-49).

We think that these evidences justify the assignment of ammonia K as a highly dispersed form of KOH adsorbate on the Fe surface, which we discuss accordingly in the manuscript text:

line 419

In one study [54], metallic K was deposited onto an Fe single crystal catalyst, resulting in a net increase in the electronic density of the iron. This enhancement was proposed to improve N_2 dissociation, explaining the higher activity of the K-promoted catalyst as compared to pure iron, and the higher concentration of dissociated N found on the K-promoted iron single crystals [56]. Although metallic K exhibited a strong promoting effect, the former studies concluded that its formation from oxidic precursors under realistic reaction conditions was unlikely. In another study [57], metallic K was found to have a stronger promotion effect than K_2O , but it was found to be volatilizing during heating and was lost from the surface. Similarly, elemental potassium was reported to be unstable on the iron surface, volatilizing during thermal treatment [58]. However, coadsorption of oxygen was shown to stabilize sub-monolayer coverages (0.10–0.30) of potassium, regardless of surface orientation. Additional single-crystal studies [55, 59] confirmed that adsorbed metallic K significantly enhanced the rate of nitrogen adsorption but desorbed under stationary reaction conditions above $\sim 670\text{ K}$. Addition of oxygen to iron pre-covered with potassium and also addition of potassium to iron pre-covered with oxygen were found to thermally stabilize the adlayer. A more recent study using supported iron nanoparticles mixed with alkaline oxide promoters (K_2O , Li_2O , Na_2O , Cs_2O) indicated that the promoters formed surface patches on the iron nanoparticles of the spent catalyst [60]. DFT simulations revealed that a stabilizing Fe–O–K interaction was required, leading to a double-layer structure with oxygen atoms forming the first sublayer and K^+ ions positioned externally. In the same line, an XPS study of K^+ ions on a mica substrate reported similar spectral features to those observed

here [34]. The authors discussed that the signal in the position of “ammonia K” may correspond to undercoordinated cations located at the surface.

line 448

Based on these arguments, we propose that “ammonia K” corresponds to a highly dispersed form of KOH adsorbate residing on the Fe surface under reaction conditions. This surface species forms in situ during activation due to hydration of K_2O . Moreover, the presence of residual moisture at levels around 10^2 ppm in industrial feeds [29] supports this assignment. The melting point of KOH ($T_m=406$ °C) suggests that at some point of the treatment it may become liquid or highly mobile, giving rise to the characteristic spectral feature observed in the XPS (Fig. 5c).

4. Figure 4h shows the changes in element distribution of the spent catalyst, but does not explain the potential relationship between these changes and ammonia production and long-term stability. Please analyze the correlation between the structural characterization results of the spent catalyst and its operational performance (such as conversion rate and selectivity).

We thank the reviewer for this insightful comment. During the reductive activation in the reaction mixture, several simultaneous processes occur:

1) reduction of the iron phase into ammonia iron, with formation of a porous architecture and the evolution of water; 2) segregation of the promoter phase; 3) chemical transformation of the promoter into the active form (ammonia K); 4) ammonia synthesis. Not all of the previously mentioned phenomena affect directly the rate of ammonia formation. However, all of them are important for the catalyst functioning. As we presented in our study, the role of the so-called textural promoters (Al, Si, and Ca to some extent) is to stimulate a tuned rate of reduction of the iron phase, such that a porous nanodispersion can be achieved. The porosity increases the reactive surface area of the catalyst and ensures a free path for the diffusion of gaseous molecules. The other role of these promoters is to stabilize such a porous architecture, as implied by the long-term stability over 10 years of this catalyst system under industrial conditions. Hence, these promoters are the reason for the long-term stability of the catalyst.

In order to make this point clearer in the revised manuscript version, we have added the following text:

Line 356

The role of these mineral phases relies on preventing agglomeration/sintering of the iron crystallites while preserving the 3D nanodispersion of the porosity (Fig. 4j,k). Hence, they also exert the templating action on the active structure and are identified here as the main reason of the structural longevity of the technical catalyst.

However, the promoter segregation leading initially to the mineral platelet and then to the crust-like aggregates at the external surface, should not be considered as the cause of the catalyst activity. It is rather a consequence of the phase immiscibility

between the in situ produced metallic iron and the oxidic promoter material. Our data support this interpretation, for instance by the fact that the transition from the platelet aggregate into the crust-like material did not affect the catalytic conversion (Fig. 2, Fig. 3j-3l, vid. 4). Hence, the presence and structure of the segregated promoter layers is not correlated to the activity.

However, the detected phases can act as a chemical protection against water poisoning, since they are very effective drying agents. We updated the text:

line 386

The setting of the resulting disordered crust-like phases (Fig. 3l) did not bring any detectable changes in the catalytic conversion (Fig. 2), suggesting that the production of ammonia was not related to the external material, but rather to the porous metallic substrate. However, the identified promoter phases are highly effective drying agents [29, 53], contributing to the chemical resilience of the catalyst and allowing it to withstand the moisture levels typically encountered in industrial processes [29].

The potassium promoter has a distinctive behavior, as shown in our XPS data of Fig. 6 and in the discussion of the previous question. At temperatures below 250°C, it is mainly found on the external surface along with the other promoters. However, at high temperature, when the ammonia synthesis is active, it is dispersed also onto the porous catalytic substrate (Fig. 7b). We think that the mobility of the active K form, the “ammonia K”, is thermally triggered. At high temperature the K promoter can move freely across the catalyst and “wets” the metallic substrate rather than the mineral platelets at the external surface. This preferential location of K, depending on the temperature is a new finding of our study.

The majority of the studies mentioned in the discussion of the K promoting effect agree with the “electronic” nature of the K promotion. For instance, the groups of Ertl (REF 2, 11, 14, 54-56, 59, 62, 63) and Somorjai (REF 25, 28, 47, 58 and 64) measured a decrease in the work function of Fe after covering it with K. We added these relevant references in the discussion to stress this point more clearly:

line 470

It also seems possible that the beneficial effect of K relies on improving the rate of nitrogen dissociation [56]. The KOH adsorbate can produce a modification on the nature and number of active sites, for instance, by increasing the local electron density of the iron surface and decreasing its work function [25, 26, 48, 54, 58].

How this change on the catalyst work function enhances the N₂ dissociation is still a matter of discussion, as it could induce direct electronic transfer from K into the N₂ antibonding orbitals, or by back-donation from the iron into the N₂, or facilitate the product desorption. We, therefore, added:

line 473

Activation of N_2 involves weakening its triple bond [7], for which electron-rich mediators such as alkaline hydroxides, nitrides, electrides, hydrides [65, 66], etc., are beneficial. The mechanism of this effect may be either by direct electronic transfer into antibonding orbitals of N_2 [7], or by an electron transfer into the iron surface that subsequently destabilizes the triple bond by π -backdonation [1, 11, 54]. The modification of the active surface can also induce higher rates of NH_3 desorption, thus preventing the self-poisoning observed at high output conditions. For instance, in situ studies using infrared spectroscopy on Fe single crystal catalysts covered with preadsorbed K found a measurable redshift in the ammonia-related frequencies [67, 68], indicative of a weakening in the ammonia–iron bond. The authors concluded that the presence of K was increasing the desorption rate of ammonia and the rate of hydrogenation from N into NH_x .

We hope that the added references and the text used to answer this and the previous questions result in a clearer picture of the promoting effect of K on this system. However, we cannot relate quantitatively the effect of K on the reaction rate from our experimental data. For this, it would be necessary to investigate catalysts with varying K contents and determine how these contents affect the catalytic rate, which is beyond the scope of our study carried out on an industrial catalyst.

As a minor clarification, we mention that our experiments in the OSEM were performed at a steady level of catalytic activity. Furthermore, after the initial activation, ammonia is the only product, hence the reaction proceeds with a 100% selectivity. In brief, there is no correlation between the promoter segregation and the selectivity, because ammonia is always the only product.

5. The synergistic effect between the distribution of nano-dispersed iron and local potassium enrichment was mentioned, but there is a lack of foundation for quantifying the catalytic conversion rate and catalytic site density. Suggest combining experimental calculations to provide measurement data on the density of surface points per unit area, to enhance the persuasiveness of the discussion.

In our previous submission, we used raw QMS data to depict the catalytic function. We were missing the calibration of the QMS lines to convert the data into actual rate numbers. We must mention that the detected conversion in the OSEM lies at the limit of what can be currently achieved with state-of-the-art techniques. It is technically very difficult to set up the reaction under clean and reproducible conditions, and due to its thermodynamic limitation, it is very hard to detect any catalytic conversion at the low-pressure regime of our experiment. We hope that in the coming years the pressure gap of operando analytics will be closed to achieve a clearer picture of the catalyst functioning under actual working conditions.

To answer the reviewer's question, we set up our system and dosed a known quantity of gaseous ammonia with a mass-flow controller, and used the QMS response to get an estimate of the conversion rate. We also searched the literature for reported conversion

rates at various conditions. We compiled this information in a new Table in the supporting information:

Table S1. Comparison between the rate of ammonia formation in the OSEM experiment, iron single crystal catalysts and industrial ammonia conditions

Catalyst type	kg(NH ₃) h ⁻¹ kg ⁻¹	mmol (NH ₃) h ⁻¹ g ⁻¹	nmol (NH ₃) s ⁻¹ cm ⁻²
Multipromoted Fe ^a	4.34 x 10 ⁻⁴	2.55 x 10 ⁻²	0.47
Fe (111) ^b	154 x 10 ⁻⁴	90.3 x 10 ⁻²	~13.1
Fe (211) ^b	114 x 10 ⁻⁴	66.9 x 10 ⁻²	~9.7
Fe (100) ^b	23.5 x 10 ⁻⁴	13.8 x 10 ⁻²	~2.0
Fe (210) ^b	21.1 x 10 ⁻⁴	12.4 x 10 ⁻²	~1.8
Multipromoted Fe ^c	755	4.42 x 10 ⁶	6.43 x 10 ⁵

a. Measured in the OSEM experiment with a 1-point calibration of the QMS signal at m/Z=15, 142.4mg of spent catalyst weight, and assuming a surface area of 15m²g⁻¹ [3].

b. Values estimated from reference [4] at 20bar and 450°C.

c. Values estimated from reference [5] at 100bar and 450°C.

The data in this table shows that our low pressure OSEM experiment is still below the activity levels measured on single-crystal iron catalysts at 20bar. We updated the text accordingly to mention the contents of this table in the manuscript:

line 165

We also calibrated the QMS signal at m/z=15 for semiquantitative determination of the ammonia conversion. As shown in Tab. S1, the estimated conversion rate was below the values reported for iron single crystals under high-pressure conditions [28].

However, it is very hard to relate these numbers to any intrinsic property of the active sites. When comparing catalysts under identical reaction conditions, the differences in activity could result from different densities of active sites, or due to different intrinsic activities of the sites.

In the examples of the table, the multi-promoted catalysts are texturally superior due to their porous structure. Hence, their surface area is much larger and can accommodate more active sites per unit mass as compared to the single crystal analogues. On the other hand, the multi-promoted catalysts also have the promoting action of the textural and electronic additives, which could imply a higher intrinsic turnover rate at the catalytic site. In practice, it is impossible to make a distinction between these possibilities without an in situ determination of the active sites under identical reaction conditions.

We think that the actual phenomenon is more complex. In (ref. 12) the authors show by molecular dynamics simulations that the active sites do not exist as a static atomic arrangement that induces the catalysis, but instead, that the catalytic sites are continuously formed and disrupted as a consequence of the chemical potential in the

course of the reaction. This transient existence practically sets the active site as an emerging property of the system. Consequently, any reliable determination of the intrinsic activity requires characterizing the active sites in situ.

Finally, we must mention that the ammonia synthesis is limited thermodynamically and kinetically. This means that many external factors such as GHSV, temperature, pressure, feed composition, trace impurities and catalyst constitution affect the reaction rate in several ways that cannot be extrapolated from one set of conditions to another. This also precludes straight comparisons of catalytic data achieved in different reactors at different conditions, as implied by the pressure disparities presented in the table.

In brief, it is impossible to elucidate any information about the intrinsic activity, i.e., the activity per active site, with the information currently at hand. However, at least qualitatively, we are satisfied that the results of our OSEM experiment are approaching the activity levels of single crystal catalysts at 20bar. In any case, these examples are far below the activity level of industrial ammonia synthesis, exemplifying the need of operando analytics under true industrial conditions.

We updated the text accordingly to highlight this in the discussion:

line 525

Finally, we mention that the activity of the multi-promoted industrial catalyst in the low-pressure OSEM experiments (Tab. S1) approaches the levels reported for single-crystal iron catalysts at 20bar, although it remains lower. However, drawing conclusions about the intrinsic activity of specific catalytic sites in any of these systems is very challenging. Variations in activity may arise from differences in the active site density or intrinsic site reactivity. This is the case not only in the as-prepared state of the pre-catalyst material, but also in its working state, where an evolution of the structure takes place. The comparison is even more complicated when considering the multi-promoted catalyst due to the influences of the additives that affect both, the texture and the electronic properties. Because the active sites are transient and form only in situ [12], in situ characterizations sensitive to the fast turnover of the sites are essential for determining the intrinsic activity. Additionally, the reaction rate in ammonia synthesis is influenced by many external factors, making a comparison of results across different conditions unreliable. Despite this, the qualitative agreement between our OSEM and high-pressure data acquired on Fe single-crystals is encouraging. The present study emphasizes the need of further developing operando characterization techniques compatible with increasingly higher pressures in order to bridge the pressure gap.

6. Is the Fe-N interaction only related to the type and distance of atoms, and how is it different from ion intercalation and adsorption in batteries (e.g. Energy Environ. Mater. 2023, 6, e12342. <https://doi.org/10.1002/eem2.12342>; Adv. Mater. 2021, 33, 2100171. <https://doi.org/10.1002/adma.202100171>)? When Fe exists in the form of ions on the surface of the catalyst, and the distance between it and N is close, what impact does it

have on the intermediate adsorption products and catalytic yield of Fe-N? By combining the aforementioned literature, the authors are encouraged to make a brief discussion with an enhanced multidisciplinary impact of their work.

In our characterization by NAP-XPS, we basically found that the iron was transformed into metallic Fe. We also found that N existed as atomic N on the iron surface during the catalytic process (Fig. 5), which is an advancement in the current knowledge of ammonia synthesis where it has been speculated that nitrides were the active species. Based on this, it seems hard to establish a link between ionic iron surface species (which we did not detect in this study during ammonia synthesis) and the intermediate fragments of ammonia formation under reaction conditions, or a correlation between the distance of the N species to the iron ions and the catalytic yield.

However, the information on the recommended references offers a very interesting approach to a generalized understanding of Fe-N interactions, where Fe may also come in variable oxidation states and N may be present as ammonium ions NH_4^+ . In the first example (Energy Environ. Mater. 2023, 6, e12342), the authors prepared an integrated catalyst for Li-S batteries composed of isolated single Fe atoms and Fe_2N nanocrystals co-decorated on nitrogen-doped graphene. The dual activity was demonstrated in lithium sulphide-based batteries, where the single atom Fe catalyzed the reduction of Li_2S_n and the Fe_2N site decomposed the Li_2S . Working together, these sites offer good possibilities in the manufacture of separator materials for Li_2S batteries. The second reference (Adv. Mater. 2021, 33, 2100171.) shows a host architecture of iron hexacyanoferrate where NH_4^+ ions can diffuse during ammonium-based battery charging/discharging for energy storage applications.

After reading the manuscripts carefully, we conclude that despite the disparity of operation conditions between ammonia synthesis and the presented examples, they all show the versatility of the Fe-N system involving a tuned interaction strength, multiple coexisting species and the dynamic adaption to changing chemical potentials.

We mention this in the text:

line 519

Hence, the N states on Fe can be better described by a kinetic loop similar to a frustrated phase transition [21, 72] involving fluctuating compositions of nitrides, subsurface species and $\text{N}+\text{NH}_x$ surface species. This behavior may explain both the long-term stability of the working catalyst and its pronounced sensitivity to poisoning. We believe that similar dynamics are also the reason to the broader versatility of Fe-N systems, which have already been harnessed in other applications, such as battery materials [73].

Reviewer #2 (Remarks to the Author):

This manuscript presents a significant advancement in the understanding of industrial ammonia synthesis catalysts, particularly multi-promoted iron-based systems. Through operando scanning electron microscopy and near-ambient pressure X-ray photoelectron spectroscopy, the authors provide good insights into the structural evolution and dynamic behavior of these Fe-based catalysts under reaction conditions. I recommend it for publication with minor revisions.

Strengths: (1) The activation process is essential for forming the active catalyst configuration. The active phase consists of a nanodispersion of metallic iron covered by mobile potassium-containing adsorbates termed "ammonia K". (2) In situ studies reveal that potassium species exhibit high mobility, redistributing between the catalyst surface and the bulk during reaction.

Suggestions: (1) For the same question, the author has explored multiple possible explanations. It is recommended that they ultimately present the interpretation they think most compelling. "For K species (Fig. 5c), the 2p_{3/2} contributions indicate a mixture of mainly K₂O (BE= 293.2 273 eV) and KOH (BE=293.8 eV) at 250°C. At 500°C, emergence of a peak at BE=294.75 eV was observed. This signal, which we refer to as "ammonia K", may indicate the formation of metallic K [30] or undercoordinated isolated surface K⁺ species [31]." "Previous studies have suggested that K at reaction conditions should exist as an anhydrous KOH adsorbate [16, 25,49] [50], because the metallic form would be volatilized." There is a potential contradiction in the interpretation of "ammonia K." Is "ammonia K" metallic K or KOH or a coordinated complex? At the end of the manuscript, the author used KOH to explain the experimental phenomenon. Could "ammonia K" be KNH₂ or K₂NH? If metallic K existed in a free state, it would likely evaporate under reaction conditions (400-500°C). However, if K is strongly interacting with the surface (Fe or oxides), it might exhibit a higher stability and remain localized.

We thank the reviewer for the time dedicated to assessing our manuscript and for the constructive feedback.

To the first question: Our data support that "ammonia K" is a highly dispersed form of KOH. We apologize that we missed the opportunity to stress this out more emphatically in the original text. We updated this part of the text to make clear that dispersed KOH is the active form, and present accordingly the experimental data, including additional references in the discussion.

To make this clear in the updated manuscript, we include that:

line 416

Based on its binding energy in the XPS spectra, the observed species could be attributed to metallic K. However, under the conditions of ammonia synthesis—where water and additional oxidic phases are present—the presence of metallic K is unlikely [16, 25, 54, 55]. In one study [54], metallic K was deposited onto an Fe single crystal catalyst, resulting in a net increase in the electronic density of the iron. This enhancement was proposed to improve N_2 dissociation, explaining the higher activity of the K-promoted catalyst as compared to pure iron, and the higher concentration of dissociated N found on the K-promoted iron single crystals [56]. Although metallic K exhibited a strong promoting effect, the former studies concluded that its formation from oxidic precursors under realistic reaction conditions was unlikely. In another study [57], metallic K was found to have a stronger promotion effect than K_2O , but it was found to be volatilizing during heating and was lost from the surface. Similarly, elemental potassium was reported to be unstable on the iron surface, volatilizing during thermal treatment [58]. However, coadsorption of oxygen was shown to stabilize sub-monolayer coverages (0.10–0.30) of potassium, regardless of surface orientation. Additional single-crystal studies [55, 59] confirmed that adsorbed metallic K significantly enhanced the rate of nitrogen adsorption but desorbed under stationary reaction conditions above ~670 K. Addition of oxygen to iron pre-covered with potassium and also addition of potassium to iron pre-covered with oxygen were found to thermally stabilize the adlayer. A more recent study using supported iron nanoparticles mixed with alkaline oxide promoters (K_2O , Li_2O , Na_2O , Cs_2O) indicated that the promoters formed surface patches on the iron nanoparticles of the spent catalyst [60]. DFT simulations revealed that a stabilizing Fe–O–K interaction was required, leading to a double-layer structure with oxygen atoms forming the first sublayer and K^+ ions positioned externally. In the same line, an XPS study of K^+ ions on a mica substrate reported similar spectral features to those observed here [34]. The authors discussed that the signal in the position of “ammonia K” may correspond to undercoordinated cations located at the surface.

line 448

Based on these arguments, we propose that “ammonia K” corresponds to a highly dispersed form of a KOH adsorbate residing on the Fe surface under reaction conditions. This surface species forms in situ during activation due to hydration of K_2O . During activation the moisture level in the feed can reach values around 10^2 ppm. [29]

Second question: The formation of potassium amide phases such as KNH_2 and K_2NH is a very interesting hypothesis that was considered as a possible cause for the K-promotion. This hypothesis was initially presented by Ozaki et al. (REF 57) in the 1970s. However, the experimental evidence collected soon after by van Ommen et al (REF. 61) basically ruled out this possibility. They investigated the equilibrium compositions of ammonia formation considering also the formation of amides at varying loadings of potassium. Furthermore, they also loaded potassium amide as a potassium source in

their experiments. In all conditions, they concluded that “no amide is formed during ammonia synthesis”.

We mention this important result in the updated version of the manuscript:

line 441

Some studies have also proposed that the potassium phase may transform into potassium amide (KNH₂) through a two-step process involving the initial formation of metallic potassium, followed by its reaction with ammonia to form the amide [57]. This hypothesis was examined through detailed equilibrium measurements under varying potassium promoter loadings [61]. Additionally, KNH₂ was introduced directly into the reactor to assess its influence on ammonia synthesis. The study concluded that neither KNH₂ nor metallic K were formed under any of the investigated conditions [61].

We further mention that there is some ppm level of moisture expected in the industrial ammonia synthesis feed, which would favor the KOH form even if any amide were originally present in the system.

(2) "It also seems possible that the beneficial effect of K relies on improving the rate of nitrogen dissociation. The KOH adsorbate can produce a modification on the nature and number of active sites, for instance, by increasing the local electron density of the iron surface." Is there any experimental basis for this statement? Please cite relevant reports.

The majority of the studies mentioned in the discussion of the K promoting effect agree with the “electronic” nature of the K promotion. For instance, the groups of Ertl (REF 2, 11, 14, 54-56, 59, 62, 63) and Somorjai (REF 25, 28, 47, 58 and 64) measured a decrease in the work function of Fe after covering it with K. We added these relevant references in the discussion to stress this point more clearly:

line 470

It also seems possible that the beneficial effect of K relies on improving the rate of nitrogen dissociation [56]. The KOH adsorbate can produce a modification on the nature and number of active sites, for instance, by increasing the local electron density of the iron surface and decreasing its work function [25, 26, 48, 54, 58].

How this change on the catalyst work function enhances the N₂ dissociation is still a matter of discussion, as it could induce direct electronic transfer from K into the N₂ antibonding orbitals, or by back-donation from the iron into the N₂, or facilitate the product desorption. We, therefore, added:

line 473

Activation of N₂ involves weakening its triple bond [7], for which electron-rich mediators such as alkaline hydroxides, nitrides, electrides, hydrides [65, 66], etc., are beneficial. The mechanism of this effect may be either by direct electronic transfer into antibonding orbitals of N₂ [7], or by an electron transfer into the iron surface that subsequently destabilizes the triple bond by π -backdonation [1, 11, 54]. The modification of the active surface can also induce higher rates of NH₃ desorption, thus preventing the self-poisoning observed at high output conditions. For instance, in situ studies using infrared spectroscopy on Fe single crystal catalysts covered with preadsorbed K found a measurable redshift in the ammonia-related frequencies [67, 68], indicative of a weakening in the ammonia–iron bond. The authors concluded that the presence of K was increasing the desorption rate of ammonia and the rate of hydrogenation from N into NH_x.

We hope that the added references and the text used to answer this and the previous question result in a clearer picture of the promoting effect of K on this system.

(3) "Our investigation is an effort to bridge the pressure and material gaps of heterogeneous catalysis, confirming that ammonia iron is not simply a particular arrangement of the α -Fe phase, but a dynamic entity that only exists in situ by the synergistic action of the various components of the technical, multi-promoted catalyst." The authors claim to bridge the pressure gap in heterogeneous catalysis; however, their operando experiments were conducted at only 22 Pa, which is far from industrial ammonia synthesis conditions (200-300 bar). I suggest the authors revise the above sentence to more accurately reflect the experimental conditions.

We agree with the reviewer that the mentioned sentence should be revised. It was not our intention to claim that we have bridged the pressure gap, but rather that our study is an attempt (still insufficient) to step into this gap. Our experimental operando conditions are certainly far off compared to the industrial reaction. However, we showed that we could reach pressure regimes where the catalyst started to become active, correlating to compositional and structural changes. We mention that our OSEM and NAP-XPS experiments lie in the border line of what is currently attainable in this context, and that they are already a significant improvement compared to what was available only a few years ago. We hope that future technical achievements will continue to serve to fill the pressure gap in the coming years.

To make this clearer, we updated the text accordingly, and now mention that:

line 559

Our investigation bridges the material gap and constitutes also an effort towards bridging the pressure gap in heterogeneous catalysis. In fact, our study highlights that ammonia iron is not simply a particular arrangement of the α -Fe phase, but a dynamic

entity that only exists in situ by the synergistic action of the various components of the technical, multi-promoted catalyst. We anticipate that further advancements in operando techniques will ultimately close the pressure gap in the coming years.

(4) About Fig.7b, if K accumulates on the surface of mineral platelet aggregate (Al-Si-Ca-Fe) rather than on the ammonia Fe surface, how does K enhance ammonia synthesis performance?

In fig. 7b, we tried to represent an “active catalyst state (500°C)”, where the K promoter is distributed over the catalytic iron substrate, and an “inactive catalyst state” where this K is found preferentially on the mineral platelet. Please note that the “active state” occurs only at high temperatures representative of ammonia synthesis (~500°C), where the ammonia K can be mobilized and redistributed across the catalyst substrate. Hence, this mobile K “wets” the metallic iron under ammonia synthesis conditions, giving rise to the promotion effect.

We think that the prior figure 7b was lacking this information. We have thus updated the figure and its caption accordingly, now including labels to mark clearly the “active” and “inactive” forms of the catalyst depending on the temperature, respectively:

Fig 7. Schematic representations of the morphological evolution of the promoter phases, the activated catalyst architecture, and the catalytic production of ammonia. *a.* The chemical linking of Al, Si, Ca, Fe oxidic phases produces a material similar to a cementitious phase. At early stages of hydration, this phase gives rise to nanofiber basic units. With ongoing aging under decreasing water vapor (TOS) the basic units assemble sequentially in platelets, needle-like aggregates and densified/disordered material. *b.* The activated catalyst exhibits a 3D nanodispersion of ammonia Fe crystallites structurally stabilized by a set of cementitious phases. Excess of the segregated promoter material accumulates at the external surface in the several

forms depicted in **a**. Ammonia K covers the ammonia Fe of the catalytic substrate under ammonia synthesis conditions (500°C), but preferentially accumulates on the external surface at low temperatures. **c**. The dissociative chemisorption of N_2 is enhanced by in situ formed ammonia K/ammonia Fe phases due to electronic transfer effects. The atomic N species reacts either sequentially with atomic H to give rise to the ammonia product, or with the Fe surface to produce nitrides. The presence of alkaline phases (KOH, $Ca(OH)_2$, CaO) increases the local alkalinity and favors the desorption of the ammonia product.